# Examining gender inequalities in factors associated with income poverty in Mexican rural households

**Juan Armando Torres Munguía**[1]*, **Inmaculada Martínez-Zarzoso**[1,2]

**1** Faculty of Economic Sciences, Georg-August-Universität Göttingen, Göttingen, Germany, **2** Department of Economics, University Jaume I., Castello de la Plana, Spain

* jtorres@gwdg.de

**Data Availability Statement:** Datasets are freely available at: https://www.coneval.org.mx/Medicion/MP/Paginas/Programas_BD_08_10_12_14_16_

## Abstract

According to 2016 official estimates, almost 60% of the rural population in Mexico (16.9 million people) had income levels below the poverty line, and approximately 29.2% (8.3 million) could not even afford the basic food basket. Whereas most poverty research disregards gender and exclusively analyzes average income or the expected probability of being poor, we depart from these approaches by examining the effect of potential risk factors on two of the lowest quantiles of income-to-poverty ratio distribution, namely the corresponding to poor and extremely poor families. Focusing on identifying heterogeneous effects according to the sex of the household head, we apply additive quantile models to a cross-sectional dataset containing information on 4,434 women-headed and 14,877 men-headed households. For each model, we introduce 45 variables at the individual/household, community, and regional levels. Two major contributions emerge from this paper. First, the identification of a subset of significant factors whose effect is independent of the head's sex and is relevant for poor and extremely poor families. This is found for the variables credit card ownership, access to basic housing services, education level, and satisfaction with public services. Second, results also identify a subset of significant factors with an uneven effect on income according to the sex of the head that is observed both in the poor and extremely poor households. Variables having this gendered effect are the community's income inequality, municipal human development, social networks, access to social security, and gender-based violence against women in the public sphere. Out of these, particularly relevant is the effect of the last three factors, whose association with income has not been explored before for rural Mexico and for which the bias among sexes increases as family income grows from extreme poverty to poverty level.

## Introduction

According to 2016 official estimates of poverty in Mexico, almost 60% of the total rural population (around 16.9 million people) had income levels below the poverty line, and approximately 29.2% (8.3 million) could not even afford the basic food basket. These facts call for the need to

18.aspx (poverty data) https://www.inegi.org.mx/programas/intercensal/2015/ (2015 Intercensal Population Survey) https://www.inegi.org.mx/programas/cngmd/2015/ (2015 National Census of Municipal and Delegation Governments) https://www.inegi.org.mx/programas/encig/2015/ (2015 National Survey of Quality and Governmental Impact) https://www.inegi.org.mx/programas/endireh/2016/ (2016 National Survey on the Dynamics of Household Relationships) https://www.datos.gob.mx/busca/dataset/declaratorias-sobre-emergencia-desastre-y-contingencia-climatologica (information on natural disasters) http://www.conapo.gob.mx/es/CONAPO/Datos_Abiertos_del_Indice_de_Marginacion (social marginalization) https://www.mx.undp.org/content/mexico/es/home/library/poverty/informe-de-desarrollo-humano-municipal-2010-2015–transformando-.html (human development index and municipal functional capacities).

**Funding:** This publication was supported financially by the Open Access Grant Program of the German Research Foundation (DFG) and the Open Access Publication Fund of the University of Göttingen. JATM received financial funding from the Göttinger Graduiertenschule Gesellschaftswissenschaften (GGG) of the University of Göttingen (https://www.uni-goettingen.de/de/29938.html). The funders had no role in study design, data collection and analysis, decision to publish, or preparation of the manuscript.

**Competing interests:** The authors have declared that no competing interests exist.

understand the associated factors of this high prevalence of rural poverty in a country that since 1994 belongs to the Organization for Economic Cooperation and Development (OECD), whose members are among the most developed and wealthiest in the world.

Income poverty in rural Mexico is not an understudied subject. Some consistent findings can be derived from previous research on the matter regarding economic, demographic, and social factors linked to this phenomenon. Broadly speaking, there is a consensus that old-age, indigenous origin, low levels of education, overcrowded families, undernutrition, community's marginalization and social deprivations are associated with higher poverty levels [1–7]. However, despite these key contributions, there are still some issues that have not been examined. On the one hand, existing studies on rural poverty mostly ignore gender, overlooking the dissimilar experiences faced by women and men in several spheres of life, such as the use of time, social networks, political and economic participation, or gender-based violence. On the other hand, most of the research is exclusively based on mean regression models analyzing the population's average income or the expected probability of being poor, disregarding specific effects in poorer income levels. We propose to fill this gap by focusing on the poorest.

Taking into account the abovementioned issues, the goal of this paper is to contribute to enhancing knowledge on rural poverty in Mexico by identifying a set of factors associated with the income-to-poverty ratio with a particular focus on detecting heterogeneous effects according to the sex of the household head. The income-to-poverty ratio, calculated by dividing the income of the household by the poverty threshold, also enables us to examine how these effects vary with the severity or depth of poverty experienced by the families. The dataset used is composed of microdata with the household as a recording unit and information on 45 explanatory factors covering three key levels of analysis. The first level includes characteristics of the individual and its household, the second level contains the community's features, and the third one incorporates features of the region of residence.

Our research methodology is based on estimating a quantile regression model that allows computing specific parameters for different intervals or subsets of the distribution of the dependent variable. Particularly, two quantiles of the income-to-poverty ratio distribution are analyzed in this paper, namely the corresponding poor and extremely poor households. Each poverty level is separately modeled for women- and men-headed households. In this way, we are able to examine the extent to which the effect of the explanatory variables on the income-to-poverty ratio differs by sex of the head and if these gendered effects are constant along the poverty spectrum. An additive structure is proposed to achieve model flexibility for capturing linear and non-linear processes [8–10]. Variable selection and model choice processes are applied by pursuing a boosting approach to obtain both model interpretability and goodness of fit [11, 12]. This method is used when dealing with high-dimensional data, even when the number of covariates exceeds the number of observations [13].

The rest of the paper is organized as follows. First, we give an overview of the theoretical and empirical approaches used. Then, we present our results focusing on the identification of gender biased effects. Posteriorly we discuss the results aiming to give some potential explanations based on previous studies and theory. Finally, we present the conclusions of this research.

## Materials and methods

### Conceptual framework

The empirical evidence concerning income poverty in rural communities indicates that poverty is linked to factors that can be classified into three main levels: individual and household characteristics, features of the community, and region-level characteristics.

With regard to individual and household characteristics, it has been extensively found that variables age, sex, level of education, marital status, access to credit, and health status are associated with poverty [14–16]. Even those findings are not consistent across countries and regions; it can be hypothesized that young people, elders, women, individuals with poor educational achievements, married persons, and those with a bad health status tend to exhibit lower income levels. From a multidimensional perspective of poverty, it can be expected that income poverty is also linked to social deprivations such as lack of access to health services, social security, and food [6, 17]. Notwithstanding its growing importance in poverty issues, the linkage of other variables such as social networks and time use has been considerably less explored. However, overall findings show that individuals with little or no social support and those dedicating more time to domestic work are expected to be poorer [18, 19].

In addition to individual and household characteristics, features at the community and regional levels are also key studying poverty. Concerning the community level, household income tends to be lower in rural settlements with poor levels of infrastructure, with lower shares of immigrant population, more vulnerable to natural disasters, marginalized, with lower shares of economic participation, and more unequal in terms of income [20–24]. At the regional level, the most common indicators found in previous research are related to the public sector and governance. On the whole, high levels of corruption, low quality of public goods and services, and weak governance are related to higher poverty levels [25–27].

Specifically analyzing the case of Mexico, prior studies have identified a set of factors whose linkage with rural poverty is widely recognized. Generally speaking, these studies have found that higher risks of rural poverty are observed in large households, in families whose head has a low educational level, indigenous, people experiencing social deprivations such as access to food or health services, and living in marginalized communities [1, 6, 28, 29]. To the best of our knowledge, there are other key factors, such as social networks, time use, corruption, and gender-based violence, whose linkage with income poverty is unexplored for rural communities in Mexico, and we aim at investigating them in this paper through a gender lens.

## Methodology and empirical strategy

**Data description and sources.** Data on household income come from the 2016 National Survey of Household Income and Expenditure (ENIGH) conducted by the National Institute of Statistics and Geography of Mexico (INEGI). In order to generate information for the dependent variable, income-to-poverty ratio, household income is then divided by the corresponding poverty threshold. The official criteria for defining poverty in Mexico is established by the National Council for the Evaluation of Social Development Policy (CONEVAL). In accordance with these criteria, a person is considered to be poor if their income is below the total cost of both the basic food basket and the non-food basket, which embraces transportation, education, health, entertainment, among others. In contrast, a person is considered extremely poor if their income is not even sufficient to cover the cost of the basic food basket [30, 31]. For rural communities, these two poverty thresholds were respectively set at MXN$ 1715.57 and MXN$ 933.20 monthly *per capita* [32]. In this way, having as reference the official poverty threshold and considering the number of family members, an income-to-poverty ratio equal to one indicates that the family is living at the poverty line. Therefore, when the ratio of income-to-poverty is less than one, the household is considered to live under poverty, and when the income-to-poverty ratio is less than 0.544 (933.20 divided by 1715.57 per household member), the household is living in extreme poverty. In other words, the income-to-poverty ratio allows us to capture how far is income of a family from the poverty line. Summary statistics of the income-to-poverty ratio are shown in the S1 Table.

The potentially associated economic, demographic, and social factors, independent variables in the model, are chosen from previous research on the matter and include characteristics at the individual, household, community, and regional levels, as described in columns 1 to 3 of Table 1. The type of relationship considered for each variable in the models is indicated in column 4 of Table 1. As shown in this column 4, given that no functional form is imposed *a priori* to continuous variables, both linear effects and nonlinearities are considered as modeling competing options for each of them. The sources for each variable are indicated in column 5 of Table 1. In addition to using the ENIGH as a source for many variables, we also incorporate information from the 2015 Intercensal Population Survey, the 2015 National Census of Municipal and Delegation Governments, the 2015 National Survey of Quality and Governmental Impact, the 2016 National Survey on the Dynamics of Household Relationships, the National Center for Prevention of Disasters (CENAPRED), the CONEVAL, the National Population Council (CONAPO) and the human development index developed by the United Nations Development Program (UNDP). More details on the datasets can be found in [31, 33–40]. Datasets are freely available at www.coneval.org.mx, www.inegi.org.mx, www.mx.undp.org and www.datos.gob.mx. After checking plausibility and omitting missing cases, the dataset corresponding to women-headed households comprises 4,434 observations, and the data on men-headed households consist of 14,877 observations.

**Model specification.** We utilize additive quantile regression models to identify how and to what extent the set of covariates is associated with the income-to-poverty ratio. Two major advantages are offered by these models in the context of this research. On the one hand, the additive modeling structure allows us to include all the potential functional forms: linear, nonlinear, interaction, random, and spatial effects concerning the factors listed in Table 1 [8–10, 13]. This is particularly key, given that instead of imposing *a priori* a specific functional shape to continuous variables, the model enables us to consider different competing effects for each one and selects the most appropriate form describing the relationship between income-to-poverty ratio and the given variable. For instance, for the variable age of the household head four alternative effects are considered, namely linear, non-linear, and interaction effects with the education level, and marital status (see Table 1). On the other hand, by computing regression parameters for two specific quantiles of the distribution of the response variable, i.e. the distribution of the income-to-poverty ratio, the quantile regression approach enables us to focus particularly on the effects of the variables on the income-to-poverty ratio of poor and extremely poor families.

To examine whether the linkages between the response variable and the independent variables vary according to the sex of the head, and according to the household income level, we estimate separately four additive models. Two models are applied to data on households headed by a woman and are estimated for the quantiles corresponding to the poor and extremely poor families. Analogously, the other two regression models correspond to poor and extremely poor man-headed families.

Formally expressing the abovementioned specifications, consider the dependent variable $y_{\tau i}^{sex}$, income-to-poverty ratio of observation $i$ at quantile $\tau$ for *sex* = {*woman*, *man*}, according to the sex of the household head, and the vectors $\mathbf{w_i} := (1, w_{i1}, \ldots, w_{ip})'$ and $\mathbf{z_i} := (z_{i1}, \ldots, z_{iq})'$ of $p$ categorical and $q$ continuous covariables. Both for women- and men-headed households, the model for the quantile $\tau$ of income-to-poverty ratio is given by:

$$y_{\tau i} = \mathbf{w}_i' \boldsymbol{\beta}_\tau + \sum\nolimits_{k=1}^{q} s_{k\tau}(\mathbf{z}_{ik}) + \varepsilon_{\tau i} \qquad (1)$$

**Table 1. List of covariates by type of effect estimated in the full model.**

| Level | Variable (type) | Definition | Relationship (type) | Source |
|---|---|---|---|---|
| **Individual / household** | Indigenous origin (*categorical*) | Indigenous self-identification of the household head.<br>**Categories:**<br>"**yes**": if the head self identifies as indigenous; and, "**no**": otherwise. | Linear | ENIGH |
| | Social networks (*categorical*) | Degree of perception of the household head on the easiness to obtain support from social networks in six hypothetical circumstances: need of money, care due to illness, help to get a job, to be accompanied to a medical appointment, collaboration to improve neighborhood conditions, and child care assistance.<br>**Categories:**<br>"**low**": if obtaining support from social networks in the majority of hypothetical situations is perceived by the head as difficult or impossible;<br>"**high**": if obtaining support from social networks in the majority of hypothetical situations is perceived by the head as easy or very easy; and,<br>"**medium**": otherwise. | Linear | CONEVAL with data from ENIGH |
| | Credit card (*categorical*) | Holding of a credit card by at least one household member.<br>**Categories:**<br>"**yes**": if at least one member holds a credit card; and, "**no**": otherwise. | Linear | ENIGH |
| | Disability (*categorical*) | Reported status of disability (having a developmental delay; a mental illness; and/or difficulties, or limitations performing one or more basic/everyday activities such as moving their arms, moving their legs, walking, seeing, hearing, speaking, toileting, eating, dressing, and/or learning basic skills or concepts) of the household head.<br>**Categories:**<br>"**yes**": if the head has a disability; and, "**no**": otherwise. | Linear | ENIGH |
| | Type of household (*categorical*) | Type of household based on the number of members, and the relationship between them.<br>**Categories:**<br>"**one-person**": consisting of only one member (head);<br>"**nuclear**": made up by the head, and their partner; the head, their partner, and their children; the head, and their children; the head, and their parents; or the head, and their siblings;<br>"**extended**": consisting of the head, their nuclear family (in case of having), and at least another member whose kinship tie with at least one of the other household members is beyond the nuclear family kinship ties (i.e. aunts, uncles, nephews, nieces, grandparents, grandchildren, and/or cousins); and,<br>"**other**": formed by the head, their nuclear family (in case of having), and/or their extended family (in case of having), and at least another member without kinship tie with any of the rest of household members. | Linear | ENIGH |
| | Access to food (*categorical*) | Reported status of the access to nutritious and quality food. The respondent is asked if in the last three months, due to lack of money or lack of other resources, at least one household member aged ≥ 18 years: had a diet based on a very small variety of foods; stopped having breakfast, lunch or dinner; ate less than he/she considers one should eat; was left without any food; felt hungry but did not eat; and/or ate just once a day or stopped eating for a whole day. Households having at least one member aged < 18 are asked the same questions to capture the information for this age group separately.<br>**Categories:**<br>"**yes**": a household without members aged < 18 is considered having access to nutritious and quality food if the respondent answered affirmatively to less than three out of the six questions made (i.e. less than three circumstances experienced in the last three months). Less than four for households having at least one member aged < 18 years.<br>"**no**": otherwise. | Linear | CONEVAL with data from ENIGH |
| | Access to health services (*categorical*) | Reported status of the access to public health services.<br>**Categories:**<br>"**yes**": if the head is ascribed or affiliated directly or by kinship to one of the public health institutions or programs; and,<br>"**no**": otherwise. | Linear | ENIGH |
| | Dwelling with adequate quality and sufficient space (*categorical*) | Reported status of the access to a dwelling with adequate quality and sufficient space. It takes into account four dwelling conditions: the floor is made of concrete or is coated; the roofs are made of concrete slab or slab joists with roof, wood, metal sheet, asbestos, or any superior quality; the walls are made of concrete, brick, block, stone, or any superior quality; and/or, if the number of household members per room (including the kitchen, but excluding hallways and bathrooms) is ≤ 2.5.<br>**Categories:**<br>"**yes**": a household is considered having a dwelling with adequate quality and sufficient space if the dwelling meets the four conditions abovementioned; and,<br>"**no**": otherwise. | Linear | CONEVAL with data from ENIGH |

(*Continued*)

**Table 1.** (Continued)

| Level | Variable (type) | Definition | Relationship (type) | Source |
|---|---|---|---|---|
| | Educational lag (*categorical*) | Reported status of the educational lag of the head. It indicates if the head is lagging behind the compulsory level of education according to their age. | Linear | CONEVAL with data from ENIGH |
| | | **Categories:** "yes": the head has an educational lag if he/she was born before 1982 and has not yet completed the elementary school level; or, if he/she was born on or after 1982 and has not yet completed the secondary level school; and, "no": otherwise. | | |
| | Access to basic housing services (*categorical*) | Reported status of the household's access to basic services. It takes into account four basic services: piped water within the dwelling (or outside, but within the dwelling grounds); drainage connected to the public service (or to a septic tank); electricity; and, use of natural or LP gas, or electricity like cooking fuel (or coal but having a chimney). | Linear | CONEVAL with data from ENIGH |
| | | **Categories:** "yes": a household is considered having access to basic services if the dwelling has access to the four services abovementioned; and, "no": otherwise. | | |
| | Access to social security (*categorical*) | Reported status of the access to social security of the head. It takes into account four circumstances: if the head is economically active and has access to social security (public health services and to the pension system); if the head is not economically active but has access to social security due to direct kinship; if the head is retired and receives a pension; and/or, if the head is ≥ 65 years old and receives a monetary transfer from a public program. | Linear | CONEVAL with data from ENIGH |
| | | **Categories:** "yes": if according to their age, working condition, and kinship, the head has access to the corresponding benefits from the social security; and, "no": otherwise. | | |
| | Education level (*categorical*) | Degree of formal education level completed by the head. | Linear; and/or interaction with age | ENIGH |
| | | **Categories:** "low": if the maximum completed level by the head is primary education; "medium": if the head has minimum secondary education and a maximum of high school; and, "high": if the head has completed at least a university degree. | | |
| | Marital status (*categorical*) | Marital status of the household head. | Linear; and/or interaction with age | ENIGH |
| | | **Categories:** "single"; "open-union"; "married"; "separated"; "divorced"; and, "widowed" | | |
| | Age | Age in years of the household head. | Linear; non-linear; interaction with education level; and/or interaction with marital status. | ENIGH |
| | Weekly housework hours | Time in hours spent on housework (washing, ironing, cooking, etc.) by the household head per week. | Linear; and/or non-linear | ENIGH |

(*Continued*)

**Table 1.** (Continued)

| Level | Variable (type) | Definition | Relationship (type) | Source |
|---|---|---|---|---|
| Community | Social marginalization (*categorical*) | Degree of social marginalization in 2015 of the municipality of household residence. It takes into account nine socioeconomic indicators at the municipal level: % of the population ≥ 15 years who are illiterate; % of the population ≥ 15 years who have not completed elementary school; % of the population living in dwellings without drainage nor toilet; % of the population living in dwellings without electricity; % of the population living in dwellings without piped water; % of the population living in overcrowding conditions (number of household members per room, including the kitchen, but excluding hallways and bathrooms, is > than 2.5); % of the population living in dwellings with dirt floor; % of the population living in settlements with < 5000 inhabitants; and, % of the employed population having an income of up to two minimum wages. The official methodology elaborated by CONAPO applies the principal component analysis to the data and reduces their dimensionality to a single variable, which is then categorized into four groups. **Categories:** **"very low"; "low"; "medium"; "high"; and, "very high"** | Linear | CONAPO |
| | Emergencies due to weather | Average annual number of declarations of emergency, disaster or contingency due to weather between 2010 and 2015 in the municipality of household residence. | Linear; and/or non-linear | CENAPRED |
| | Gini index | Gini index in 2015 of the municipality of household residence. | Linear; and/or non-linear | CONEVAL |
| | Human development index | Human development index in 2015 of the municipality of household residence. | Linear; and/or non-linear | PNUD |
| | Municipal functional capacities | Local functional capacities index in 2015 of the municipality of household residence. This is a composite indicator taking into account five functional capacities of the municipal public administration: capacity to involve relevant stakeholders; capacity to diagnose; capacity to formulate public policies and strategies; capacity to budget, manage, and implement; and, capacity to evaluate. | Linear; and/or non-linear | PNUD |
| | Women-to-men ratio of housework hours | Number of hours spent by women aged ≥ 12 years doing housework per hour spent by men aged ≥ 12 doing housework in 2015 in the municipality of household residence. | Linear; and/or non-linear | ENIGH |
| | Women's political participation | % of senior positions in the local public administration held by women in 2015 in the municipality of household residence. Expressed in decimal form. | Linear; and/or non-linear | National Census of Municipal and Delegational Governments |
| | Migration of women | % of the 2015 women's population aged ≥ 5 years in the municipality of household residence who lived in another state or country in 2010. Expressed in decimal form. | Linear; and/or non-linear | Intercensal Population Survey |
| | Migration of men | % of the 2015 men's population aged ≥ 5 years in the municipality of household residence who lived in another state or country in 2010. Expressed in decimal form. | Linear; and/or non-linear | Intercensal Population Survey |
| | Women's household headship | % of the 2015 population living in women-headed households in the municipality of household residence. | Linear; and/or non-linear | Intercensal Population Survey |
| | Women's economically active population | % of the 2015 women's population aged ≥ 12 years who were economically active in the municipality of household residence. Expressed in decimal form. | Linear; and/or non-linear | Intercensal Population Survey |
| | Men's economically active population | % of the 2015 men's population aged ≥ 12 years who were economically active in the municipality of household residence. Expressed in decimal form. | Linear; and/or non-linear | Intercensal Population Survey |
| | Women working in the primary sector | % of the 2015 women's working population aged ≥ 12 years who were employed in the primary sector in the municipality of household residence. Expressed in decimal form. | Linear; and/or non-linear | Intercensal Population Survey |
| | Men working in the primary sector | % of the 2015 men's working population aged ≥ 12 years who were employed in the primary sector in the municipality of household residence. Expressed in decimal form. | Linear; and/or non-linear | Intercensal Population Survey |
| | Women working in the secondary sector | % of the 2015 women's working population aged ≥ 12 years who were employed in the secondary sector in the municipality of household residence. Expressed in decimal form. | Linear; and/or non-linear | Intercensal Population Survey |
| | Men working in the secondary sector | % of the 2015 men's working population aged ≥ 12 years who were employed in the secondary sector in the municipality of household residence. Expressed in decimal form. | Linear; and/or non-linear | Intercensal Population Survey |
| | Women working in the trade sector | % of the 2015 women's working population aged ≥ 12 years who were employed in the trade sector in the municipality of household residence. Expressed in decimal form. | Linear; and/or non-linear | Intercensal Population Survey |
| | Men working in the trade sector | % of the 2015 men's working population aged ≥ 12 years who were employed in the trade sector in the municipality of household residence. Expressed in decimal form. | Linear; and/or non-linear | Intercensal Population Survey |
| | Women working in the service sector | % of the 2015 women's working population aged ≥ 12 years who were employed in the service sector in the municipality of household residence. Expressed in decimal form. | Linear; and/or non-linear | Intercensal Population Survey |
| | Men working in the service sector | % of the 2015 men's working population aged ≥ 12 years who were employed in the service sector in the municipality of household residence. Expressed in decimal form. | Linear; and/or non-linear | Intercensal Population Survey |
| | Municipality of residence (*categorical*) | Municipality of household residence. | Random | ENIGH |
| | Centroid coordinates | Longitude and latitude of the centroid of the municipality of household residence. | Spatial | INEGI |

(*Continued*)

Table 1. (Continued)

| Level | Variable (type) | Definition | Relationship (type) | Source |
|---|---|---|---|---|
| Region | Corruption | % of the 2015 population aged $\geq$ 18 years who considered corruption a common or very common problem in their region of residence. Expressed in decimal form. | Linear; and/or non-linear | National Survey of Quality and Governmental Impact |
| | Satisfaction with public services | % of the 2015 population aged $\geq$ 18 years who were satisfied with the basic and on-demand public services provided in their region. Expressed in decimal form. | Linear; and/or non-linear | National Survey of Quality and Governmental Impact |
| | Gender-based violence against women and girls at school | % of the 2016 women's population aged $\geq$ 15 years who were victims of psychological, physical, and/or sexual gender-based violence at school between October 2015 and October 2016 in the region of household residence. Expressed in decimal form. | Linear; and/or non-linear | National Survey on the Dynamics of Household Relationships |
| | Gender-based violence against women and girls in the workplace | % of the 2016 women's population aged $\geq$ 15 years who were victims of psychological, physical, and/or sexual gender-based violence in the workplace between October 2015 and October 2016 in the region of household residence. Expressed in decimal form. | Linear; and/or non-linear | National Survey on the Dynamics of Household Relationships |
| | Gender-based violence against women and girls in the family context | % of the 2016 women's population aged $\geq$ 15 years who were victims of economic, psychological, physical, and/or sexual gender-based violence in the family context between October 2015 and October 2016 in the region of household residence. Expressed in decimal form. | Linear; and/or non-linear | National Survey on the Dynamics of Household Relationships |
| | Gender-based violence against women and girls by an intimate partner | % of the 2016 women's population aged $\geq$ 15 years who were victims of economic, psychological, physical, and/or sexual gender-based violence by an intimate partner between October 2015 and October 2016 in the region of household residence. Expressed in decimal form. | Linear; and/or non-linear | National Survey on the Dynamics of Household Relationships |
| | Gender-based violence against women and girls in the public sphere | % of the 2016 women's population aged $\geq$ 15 years who were victims of psychological, physical, and/or sexual gender-based violence in the public sphere (perpetrated by a friend, an acquaintance or a stranger with whom the victim has no family nor intimate relationship, the perpetrator is not her co-worker nor her schoolmate) between October 2015 and October 2016 in the region of household residence. Expressed in decimal form. | Linear; and/or non-linear | National Survey on the Dynamics of Household Relationships |
| | State of residence (*categorical*) | Region of household residence. | Random | ENIGH |

Note: See the corresponding source (last column) for more details on the methodology. The variables not classified as (categorical) in column 2 are continuous. Summary statistics are shown in S2–S5 Tables.

Introducing the independent variables from Table 1 into model (1), the full model can be expressed as:

$$y_{\tau i} = \boldsymbol{\beta}_{0\tau} + \sum_{j=1}^{14} \mathbf{w}'_{ij}\boldsymbol{\beta}_{j\tau} + \sum_{k=1}^{28} s_{k\tau}(\mathbf{z}_{ik}) + \sum_{l=13}^{14} \delta_{l\tau}(\boldsymbol{z}_{i28}, \boldsymbol{w}_{il}) + \sum_{s=1}^{2} \vartheta_{s\tau}(\boldsymbol{rn}_s) + \varphi_\tau(\boldsymbol{sp}_i) + \boldsymbol{\varepsilon}_{\tau i} \quad (2)$$

where $\boldsymbol{\beta}_{0\tau}$ is the quantile-specific model intercept, and $\epsilon_{\tau i}$ represents the quantile-specific regression errors. The other five right-hand-side terms in Eq (2) are described below.

1. $\sum_{j=1}^{14} \mathbf{w}'_{ij}\boldsymbol{\beta}_{j\tau}$ is the parametric component for estimating linear effects of the 14 categorical variables included in Table 1.

2. $\sum_{k=1}^{28} s_{k\tau}(\mathbf{z}_{ik})$ is the model component for the 28 univariate continuous variables included in Table 1, where parameters $s_{k\tau}(\mathbf{z}_{ik})$ are smooth functions [41]. In turn, each of these functions can be decomposed into a linear part and a non-linear polynomial estimated by P-splines [41]. This decomposition is key in this paper, since it enables us to leave *a priori* the functional form of the relationship between income-to-poverty ratio and the continuous covariates unspecified and selected during the estimation process. As a consequence, the effect of every $s_{k\tau}(\mathbf{z}_{ik})$ can result in:

   a. Non-significant effect;

   b. 'Purely' linear effect;

   c. Non-linear effect; or,

   d. A combined effect of a linear and a non-linear effects.

3. The component for interaction effects is denoted by $\sum_{l=13}^{14} \delta_{l\tau}(\boldsymbol{z}_{i28}, \boldsymbol{w}_{il})$. In particular, it allows us to capture how the association between income-to-poverty ratio and age of the head varies according to their education level and with their marital status.

4. The function $\sum_{s=1}^{2} \vartheta_{s\tau}(\boldsymbol{rn}_s)$ represents random effects for the unobserved heterogeneity across municipalities and states, respectively.

5. Spatial effects are introduced in $\varphi_\tau(\boldsymbol{sp}_i)$, estimated by a bivariate tensor product P-splines [11].

Given the complex structure of the model in Eq (1) and its high dimensionality, an estimation cannot be computed by traditional methods. To overcome this issue, we implemented the following three steps.

**Step 1: Estimation via component-wise gradient boosting algorithm**. This method is a computer-intensive iterative process that combines estimation with automatic identification of significant covariates (variable selection) and determination of the functional form of their linkage with the dependent variable (model choice) [42, 43]. For each of the models estimated in this paper, 5000 initial boosting iterations are performed. Cross-validation is used to prevent overfitting resulting from running this algorithm until convergence and for finding the finite number of iterations, optimizing the prediction accuracy. By doing this, multi-collinearity problems are avoided [42].

**Step 2: Stability selection**. Once the model is fitted at the optimal number of iterations in step 1, we execute stability selection, as proposed by [44], to avoid the erroneous selection of

non-relevant variables. By using subsampling procedures, this method simulates a finite number of random subsets of the data, and then, in each of these subsets, it controls the error rate for the numcber of falsely selected noise variables while selecting relevant variables in the fitting process of the boosting algorithm. After this finite number of subsets have been fitted, the relative selection frequency per covariate effect is determined by calculating the proportion of subsets for which an effect is selected as relevant. All the effects with a relative frequency of selection equal or greater than a threshold previously specified are declared as stable effects. As a result of this selection, a parsimonious model is derived consisting exclusively of stable factors, i.e. we obtain a model with only non-zero regression coefficients. Regression coefficients for factors that are not selected as stable equal zero, indicating that they have no influence on the response variable. Setting these coefficients to zero is key, since it enables the variable selection and model choice processes. In this paper, we use 50 subsampling replicates and a threshold for the relative selection frequency of 0.8, i.e. for a covariate effect to be considered stable, it has to be selected as an influential predictor in at least 80% of the 50 random subsets. See [44] and [45] for details.

**Step 3: Pointwise bootstrap confidence intervals**. Finally, 95% confidence intervals for the subset of effects selected as stable in step 2 are calculated by drawing 1000 random samples from the empirical distribution of the data using a bootstrap approach based on pointwise quantiles [46]. In this way, a stable effect is found significant if its corresponding 95% confidence interval does not contain zero. All computations are implemented in the R package "mboost" [43].

## Results

Table 2 shows the coefficients of the relevant factors for women- and men-headed households living either in poverty or in extreme poverty. These coefficients indicate the effect of each factor on the income-to-poverty ratio. It is important to keep in mind that this ratio measures how far or close is a family to live in poverty based on their income, and therefore coefficients can be interpreted as estimates of the size of the covariate effect as a proportion of the poverty line, i.e. as a share of the income required to cover the cost of the household basic food basket and the non-food basket. Interpretation of the coefficients in the quantile regression model context is basically the same as in other traditional approaches. For categorical covariates, parameters indicate the difference in the estimated effect of a category on the income-to-poverty ratio with respect to the corresponding effect of the reference category. For example, when examining the estimated coefficients of women-headed households living in extreme poverty, results indicate that families without access to credit cards have an income-to-poverty ratio that is smaller by 0.114 units in comparison to their counterparts having credit cards. For continuous covariates with purely linear effects, the parameter indicates the change in the income-to-poverty ratio per unit change in the continuous covariate. For instance, for men-headed households living in extreme poverty, an increase of one year in the household head's age decreases the income-to-poverty ratio by 0.003 units. For continuous covariates with non-linear effects, interpretation is best done by visualizing the corresponding figures. A comparison of the estimated coefficients from models provides a clear picture of how the effect of the covariates varies across the poverty spectrum and according to the sex of the head.

Overall, the subset of relevant effects refers to 17 variables selected as significant in at least one of the four estimated models. At the individual and household level, these are social networks, credit card ownership, type of household, access to food, educational lag, access to

**Table 2. Summary of estimated coefficients for covariates with stable significant effects and their 95% confidence intervals (CI).**

| Variable | Category | Extremely poor | | | | Poor | | | |
|---|---|---|---|---|---|---|---|---|---|
| | | Women-headed | | Men-headed | | Women-headed | | Men-headed | |
| | | Coef. | 95% CI | Coef. | 95% CI | Coef. | 95% CI | Coef. | 95% CI |
| Social networks (reference category: low) | medium | | | | | | | | |
| | high | | | 0.052 | [0.036, 0.067] | | | 0.093 | [0.07, 0.116] |
| Credit card (reference category: yes) | no | -0.114 | [-0.165, -0.05] | **-0.13** | **[-0.158, -0.11]** | -0.224 | [-0.29, -0.163] | **-0.25** | **[-0.287, -0.21]** |
| Type of household (reference category: nuclear) | one-person | | | | | 0.066 | [0.027, 0.114] | 0.667 | [0.573, 0.770] |
| | extended | | | | | -0.11 | [-0.149, -0.07] | | |
| | other | | | | | | | | |
| Access to food (reference category: no) | yes | **0.099** | **[0.057, 0.139]** | 0.075 | [0.061, 0.090] | | | | |
| Educational lag (reference category: yes) | no | | | 0.033 | [0.019, 0.048] | **0.094** | **[0.043, 0.144]** | 0.051 | [0.027, 0.075] |
| Access to basic housing services (reference category: no) | yes | 0.057 | [0.023, 0.093] | **0.055** | **[0.04, 0.072]** | 0.111 | [0.067, 0.154] | **0.12** | **[0.097, 0.141]** |
| Access to social security (reference category: no) | yes | 0.142 | [0.096, 0.187] | **0.225** | **[0.2, 0.251]** | 0.159 | [0.105, 0.215] | **0.31** | **[0.275, 0.344]** |
| Education level (reference category: low) | medium | | | | | | | | |
| | high | 0.594 | [0.222, 1.002] | **0.319** | **[0.261, 0.415]** | 0.917 | [0.567, 1.825] | **0.748** | **[0.602, 0.911]** |
| Marital status (reference category: single) | married | | | | | | | | |
| | separated | | | | | | | **0.299** | **[0.191, 0.404]** |
| | divorced | | | | | | | | |
| | widowed | | | | | | | | |
| | open union | | | | | | | | |
| Age | | | | **Linear, coef.: -0.003** | | | | | |
| Age by education level | low | | | | | | | | |
| | medium | **Non-linear, inverted U-shaped curve** | | | | | | | |
| | high | | | | | | | | |
| Weekly housework hours | | | | | | | | **Non-linear, inverted U-shaped curve** | |
| Gini index | | Linear, coef.: -1.26 | | Linear, coef.: -1.08 | | Linear, coef.: -1.89 | | Linear, coef.: -1.19 | |
| Human development index | | Linear, coef.: 0.49 | | Linear, coef.: 0.9 | | Linear, coef.: 0.75 | | Linear, coef.: 1.8 | |
| Women's household headship | | **Linear, coef.: 0.65** | | | | | | | |
| Women's economically active population | | Linear, coef.: 0.33 | | **Linear, coef.: 0.26** | | Linear, coef.: 0.42 | | | |
| Satisfaction with public services | | **Linear, coef.: 0.22** | | **Linear, coef.: 0.29** | | **Linear, coef.: 0.75** | | **Linear, coef.: 0.64** | |
| Gender-based violence against women and girls in the public sphere | | | | **Linear, coef.: 0.41** | | | | **Linear, coef.: 0.56** | |

Notes: Results in **bold** letters indicate that the covariate effect on the response variable varies with household income level, i.e. keeping the sex of the family head fixed, the 95% confidence intervals of the estimated coefficients for the extremely poor and the poor households do not overlap. Similarly, results highlighted in grey indicate that the covariate effect on the response variable varies according to the sex of the household head, i.e. keeping poverty level fixed, the 95% confidence intervals of the estimated coefficients for the women- and men-headed households do not overlap. One thousand random samples from the empirical distribution of the data are used to compute the confidence intervals. For categorical covariates, parameters indicate the difference in the estimated coefficient for a category with respect to the reference category coefficient, shown in parenthesis under the name of the covariate. For continuous covariates with linear effects, only the coefficient of the mean is reported; see [46] for more details on pointwise bootstrap confidence intervals. Empty cells indicate that the corresponding factor is not selected as stable for that specific model, and therefore their coefficient is set to zero. See Table 1 for definitions of variables. See S6 Table for a summary of estimated coefficients in terms of standard deviations.

basic housing services, access to social security, education level, marital status, age, and weekly housework hours. At the community level, four covariates are selected, namely the Gini index as a proxy for income inequality, the human development index, women's household headship, and women's economically active population. Finally, two variables describing the region's characteristics are also found to be relevant: satisfaction with public services, and gender-based violence against women and girls in the public sphere. Linear, non-linear, and interaction effects are selected as functional shapes to describe these associations most appropriately. The rest of the variables are not found to be associated with income-to-poverty ratio in any of the models.

In the following lines we comment in detail on the results reported in Table 2. In some cases, to provide more details about the size of the estimated coefficients we compare them with their corresponding standard deviation (see S6 Table). We classify the findings into two groups: risk factors whose association with the income-to-poverty ratio do not vary according to the sex of the household head (no gender bias) and those having heterogeneous effects between women- and men-headed households (gender bias). In turn, these two groups of risk factors can be divided into those whose estimated effects do not differ among poverty levels and into those risk factors with an income-poverty-level varying effect.

## Income poverty risk factors without gender bias

We find no significant differences between the coefficient estimated for women-headed households and the corresponding one for men-headed households within the same poverty level in seven variables with effects selected as relevant in at least one of the four models. These variables are credit card ownership, access to food, educational lag, access to basic housing services, education level, economically active women population, and satisfaction with public services. It is important to remark, that for some variables the homogeneous effect between sexes is exclusively found in one of the two levels of poverty, but in other cases it is constantly observed for both poverty levels.

The results indicate that having a household member that holds a credit card is consistently linked to a greater income-to-poverty ratio in rural families. For extremely poor families, not having access to credit cards reduces their income-to-poverty ratio by 0.114 units (around 2.85 times their standard deviation) in women-headed households, and by 0.13 units (more than three standard deviations) for their male-headed counterparts. The magnitude of the effect significantly varies across poverty levels, but only for men-headed households (see confidence intervals in Table 2). The estimated parameter for the effect of not holding a credit card on the ratio of income-to-poverty of poor families is -0.224 (2.39 standard deviations) for women-headed households and -0.25 (2.55 standard deviations) for those headed by a man.

Concerning access to nutritious and quality food, it is relevant only for families living in extreme poverty. Broadly speaking, results suggest that extremely poor households with access to food have a greater income-to-poverty ratio than extremely poor households deprived of food. This association is 0.099 (2.47 standard deviations) for families headed by a woman, and 0.075 (1.78 times their standard deviation) for men-headed households.

Results also indicate that families whose head is lagging behind the compulsory level of education are expected to show a lower income level compared to households whose head has no educational lag. Only for households living in poverty, no evidence on gender-biased effects is found. For poor families, the estimated parameter for households with a head not lagging the compulsory education is 0.094 (one standard deviation) for women-headed families, and 0.051 (half standard deviation) for households headed by a man.

Having a house accessing to basic services is positively associated with income in rural households. The estimated parameters do not differ between women and men-headed households when keeping the poverty level constant. For the models estimated for the quantile corresponding to extreme poverty, the effects are estimated at 0.057 (1.42 standard deviations) and 0.055 (1.3 standard deviations) for households with a woman as head and those with a man as head, respectively. Poor households living in a house with access to basic services have an income-to-poverty ratio greater in about 0.111 (1.19 times their standard deviation) for women-headed households and 0.12 (1.22 standard deviations) for men-headed families than households deprived of basic housing services. Only for men-headed households, differences between levels of poverty are observed.

Moreover, in contrast to low and medium levels of education, having a high level of education (at least a university degree) is associated with a higher household income level in rural Mexico. For families in extreme poverty, the estimated parameter for those having a highly educated woman as the head is 0.594 (more than 14.85 times their standard deviation), and 0.319 (equal to 7.55 standard deviation) for families with a highly educated man as the head. As one moves up the quantile distribution to the poverty level, dissimilarities are observed but only for men-headed households. The coefficients for the poor households are 0.917 (9.8 standard deviations) for families with a woman as head and 0.748 (7.63 times in terms of standard deviations) for those headed by a man.

Concerning women's economically active population in the community of residence, results indicate that it is positively associated with the income-to-poverty ratio for women- and men-headed households in extreme poverty and for poor households headed by a woman. No significant gender differences are found for extremely poor households. As can be seen in Fig 1, for extremely poor families, the 95% confidence intervals of the estimated coefficients for the women- and men-headed households completely overlap. A one-percent rise in the share of women involved in the economic activity of the community is associated with an

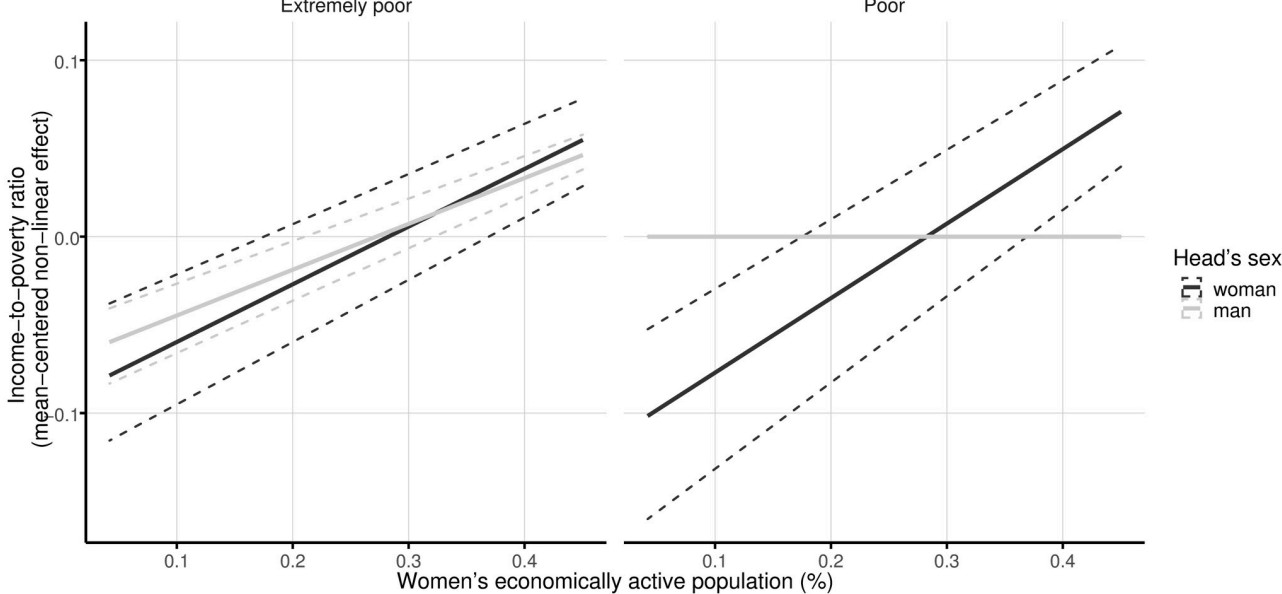

**Fig 1. Linear effects of women's economically active population on the income-to-poverty ratio by sex of the head and poverty level.** The solid lines represent the mean effects, and the dashed lines indicate 95% confidence intervals. One thousand random samples from the empirical distribution of the data are used to compute the confidence intervals.

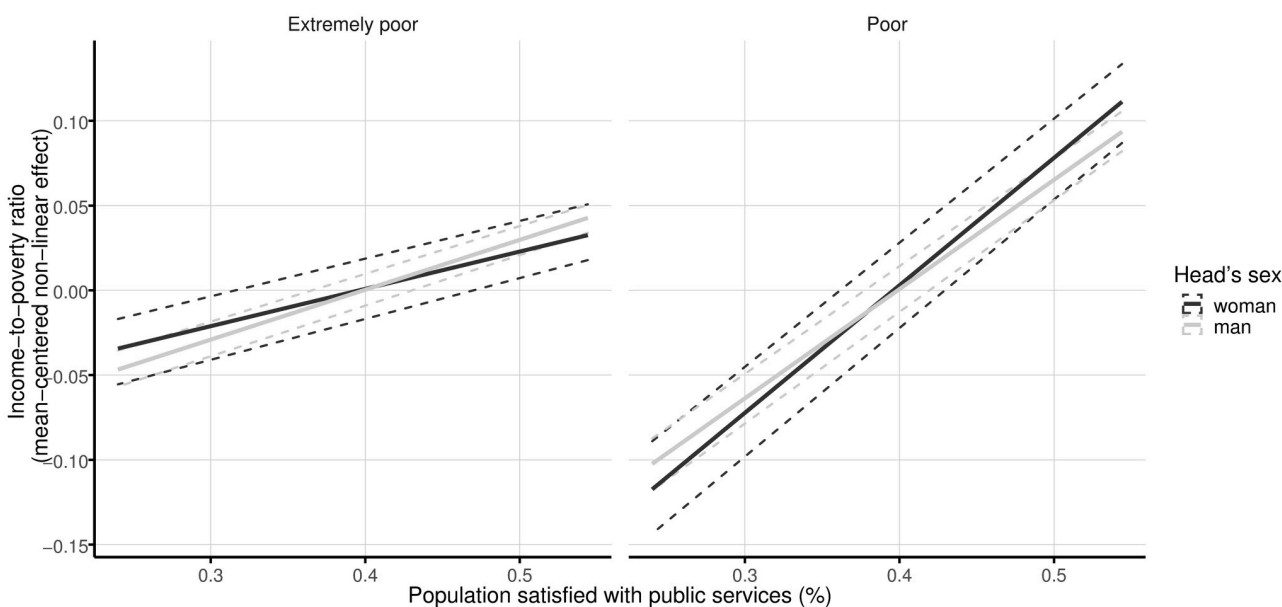

**Fig 2. Linear effects of satisfaction with public services on the income-to-poverty ratio by sex of the head and poverty level.** The solid lines represent the mean effects, and the dashed lines indicate 95% confidence intervals. One thousand random samples from the empirical distribution of the data are used to compute the confidence intervals.

increase of 0.0033 in the income-to-poverty ratio of extremely poor families headed by a woman. The estimated effect of a one-percent increase in the percentage of women economically active on the income-to-poverty ratio of men-headed households in extreme poverty is 0.0026.

Finally, regarding satisfaction with public services provided in the region of residence, a positive relationship is found between this variable and the ratio of income-to-poverty of households in rural Mexico (see Fig 2). This association is selected as stable and significant in the four models. For extremely poor families, the parameter is estimated at 0.22 (5.5 standard deviations) for households having a woman as the head and at 0.29 (6.86 times its standard deviation) for households with a man as the head. For poor households, the association is 0.75 (slightly greater than eight standard deviations) for women-headed households and 0.64 (equal to 6.53 standard deviations) for men-headed households. As shown in Fig 2, both for poor and extremely poor households, the confidence intervals for the women- and men-headed completely overlap.

## Income poverty risk factors with gender bias

As observed in Table 2, a total of 13 covariates are found to have significant gendered effects in at least one of the models estimated, i.e. keeping poverty level unchanged, confidence intervals of the parameters estimated for the women- and men-headed households do not intersect. For the variables educational lag, age, and the interaction of age with medium level of education, unequal gender effects are observed only for extremely poor families. At this poverty level, the association between education lag and the ratio of income-to-poverty is only significant for men-headed families. This parameter is 0.033 (0.78 standard deviations) for households with a head not lagging behind the compulsory education level. Age has a significant effect only for men-headed households in extreme poverty. In particular, as the age of the head increases by

one year, the household income-to-poverty ratio of these rural families sinks by approximately 0.003 units. As shown in Table 1, interaction effects of education level and age of the head are considered as a modeling alternative. In this regard, only the age varying effect of families with a woman as head having a medium level of education is selected as relevant, and an inverted U-shaped curve describes its correlation. This means that these rural households experience lower income levels in the youngest and oldest ends of the age spectrum, reaching a maximum between approximately 50 and 60 years (see Fig 3).

Variables with gender effects found to be significant only for the poor are the type of household, marital status, weekly housework hours, and women's economically active population in the community. In particular, in rural communities in Mexico, greater income-to-poverty ratios are expected in one-person households. For women-headed households in poverty, the expected income-to-poverty ratio of one-person households is greater in approximately 0.18 units (distance between their parameters, 0.066 and -0.11) in comparison to extended households (those composed of a nuclear family group and other family members, such as aunts, uncles, grandparents, cousins, etc.). Moreover, it is greater in 0.066 units (slightly greater than 0.7 standard deviations) in comparison to nuclear families and other household structures. For

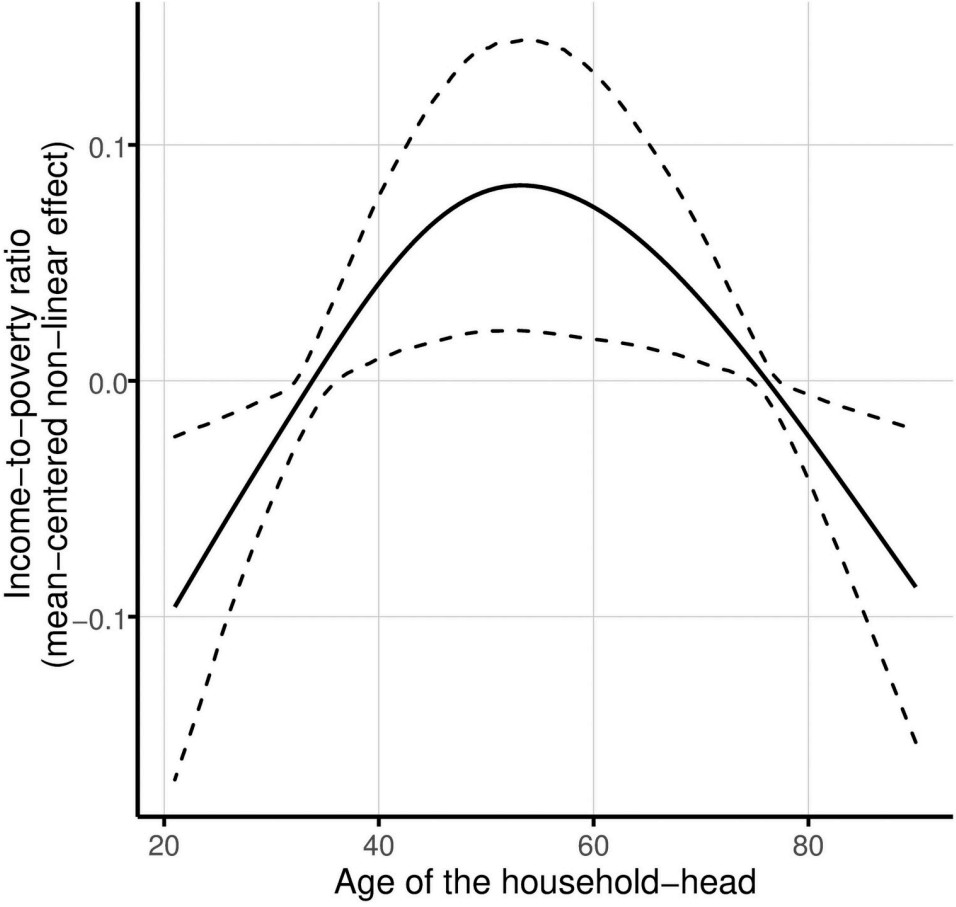

**Fig 3. Non-linear age-varying effects of education on the income-to-poverty ratio for extremely poor rural households headed by a woman with a medium level of education.** The solid line represents the mean effect, and the dashed lines indicate 95% confidence intervals. One thousand random samples from the empirical distribution of the data are used to compute the confidence intervals. As indicated in Table 1, the medium level of education specifies that the head has a minimum of secondary education and a maximum of high school level education.

poor households headed by a man, the difference between the income-to-poverty ratio for one-person families and the rest of household types is approximately 0.667 (equal to 6.8 standard deviations), which is almost ten times the corresponding parameter estimated for their women-headed counterparts. Regarding marital status, it is selected as influential only for men-headed households living in poverty. Specifically, families with a separated man as head show a greater income-to-poverty ratio. The coefficient of this linkage is 0.299 (3.05 times its standard deviation). At the individual / household level, the linkage between income-to-poverty ratio and weekly hours doing housework is selected as relevant only in the model for men-headed households living at the poverty line (see Fig 4). For these families, the linkage is represented by an inverted U-shaped curve indicating that households whose head spends less than 5 hours a week or more than 20 are associated with lower income levels. About the effect of women's economically active population in the community of residence, results suggest a gender effect indicating a positive association with the income-to-poverty ratio for women-headed poor households.

In addition, we also identify a subset of factors with an uneven effect on income according to the head's sex observed both in the poor and extremely poor households. Variables having this gendered effect are social networks, access to social security, Gini index, human

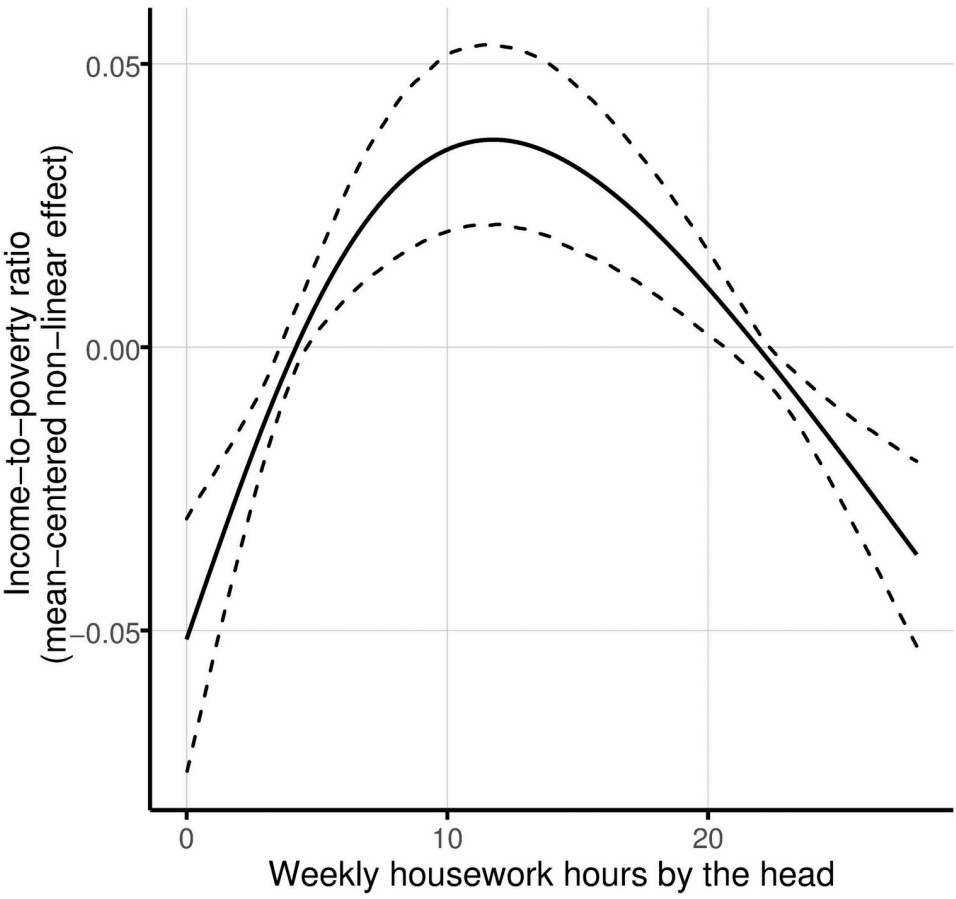

**Fig 4. Non-linear effects of weekly housework hours by the head on the income-to-poverty ratio for poor rural households headed by a man.** The solid line represents the mean effect, and the dashed lines indicate 95% confidence intervals. One thousand random samples from the empirical distribution of the data are used to compute the confidence intervals.

development, and gender-based violence against women in the public sphere. About the perception of social networks, contrary to the observed in families with a woman as head, it is found that for men-headed households having a high degree of connectedness with other people is linked to a greater income-to-poverty ratio compared to families whose head has a medium or low degree of social networks. It is key to highlight how this effect varies with income level. For extremely poor families, the estimated coefficient is 0.052 (equal to 1.23 standard deviations), whereas the effect is significantly greater for poor households, 0.093 (0.95 standard deviations).

Findings also indicate that access to social security is linked to a greater income-to-poverty ratio. For extremely poor families, the parameters are 0.142 (3.55 standard deviations) for women-headed households and 0.225 (5.33 standard deviations) for men-headed families. This effect is greater for households living in poverty, whose estimated parameters are 0.159 (equal to 1.7 standard deviations) for households headed by a woman and 0.31 (3.16 standard deviations) for households headed by a man. The Gini index of the community of residence is selected as a relevant variable in all four models (see Fig 5). In general, as income inequality in the municipality decreases, the household income-to-poverty ratio goes up. For extremely poor households, this association is estimated at -1.26 and -1.08, respectively, for women- and men-headed households. For poor households, the corresponding effect for families headed by a woman is -1.89, and for those headed by a man, the coefficient is -1.19. Even if all the parameters point to a negative association with income poverty, gender differences are observed in communities with the lowest levels of income inequality, in which the effect for women-headed households is expected to be larger in comparison to households headed by a man.

Moreover, the human development index of the community is also found to be stable and significant for all four groups considered. However, contrary to the linkage found between the Gini index and the income-to-poverty ratio, the estimated parameter for the association of the human development index in the four models is linearly increasing (see Fig 6). Specifically, for

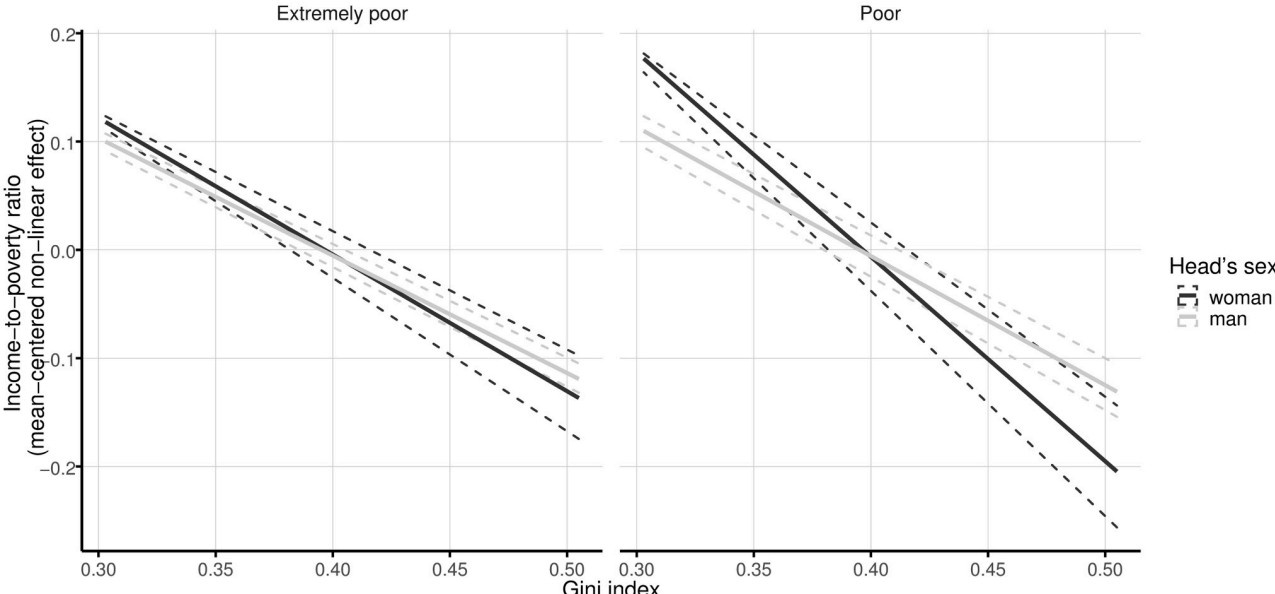

**Fig 5. Linear effects of Gini index on the income-to-poverty ratio by sex of the head and poverty level.** The solid lines represent the mean effects, and the dashed lines indicate 95% confidence intervals. One thousand random samples from the empirical distribution of the data are used to compute the confidence intervals.

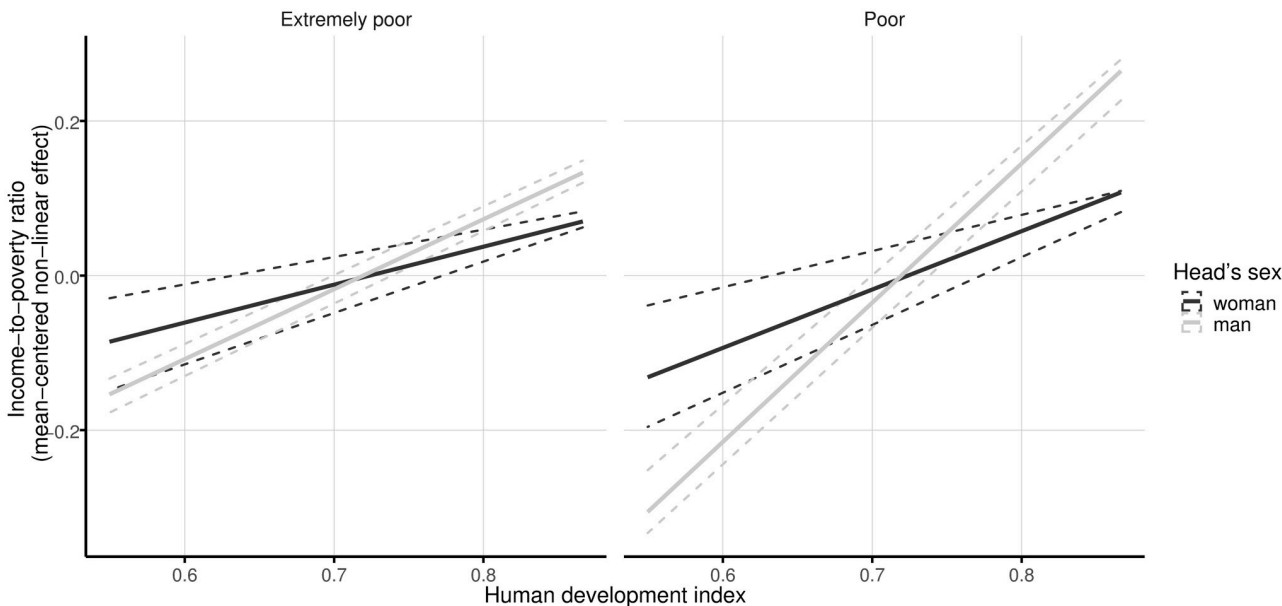

**Fig 6. Linear effects of human development index on the income-to-poverty ratio by sex of the head and poverty level.** The solid lines represent the mean effects, and the dashed lines indicate 95% confidence intervals. One thousand random samples from the empirical distribution of the data are used to compute the confidence intervals.

families living in extreme poverty, the coefficient for those headed by a woman is 0.49, and for the households headed by a man the coefficient is 0.9. For households living in poverty, in those whose head is a woman the coefficient is 0.75, and for the families with a man as head the association is estimated at 1.8 (see Table 2). Gender differences against women-headed families are observed in municipalities with the highest indexes of human development. In particular, there are also gender inequalities in the communities with the lowest levels of human development for poor households.

Women's household headship at the community level only shows a significant effect in the model for women-headed rural households living in extreme poverty. In particular, an increase of one percent in the share of people living in women-headed households is associated with an improvement of 0.0065 in the family's income-to-poverty ratio (see Table 2). At the regional level, the variable gender-based violence against women and girls in the state of residence is stable and significant only for the income-to-poverty ratio of men-headed households. Specifically, an increase of one percent in the percentage of the women's population who was a victim of gender-based violence in the public sphere (perpetrated by a friend, an acquaintance or a stranger with whom the victim has no family nor intimate relationship, the perpetrator is not her co-worker nor her schoolmate) in the last 12 months is associated with a rise of 0.0041 in the household income-to-poverty ratio for men-headed families. This effect is greater for households headed by a man living in poverty, and it is estimated at about 0.0056 for a growth of one percent in the covariate.

## Discussion

All the risk factors presented in the previous section imply statistical relationships between the selected variables and the household income-to-poverty ratio, and even though they do not necessarily imply causality, they provide evidence about key aspects for the studies on poverty

in rural Mexican households, and some potential explanations can be derived based on existing studies and theories.

As previous research on poverty has found, social networks can help people get a job, provide financial assistance, help in childcare, influence economic decisions such as investment or expenditures, and impact on income [47]. It is important to note that a lack of social connectedness may be an effect of poverty and a consequence. Just as the social, emotional, and financial support from close friends works as a strategy for coping with poverty, lower income levels also reduce the possibility of socializing. It is of special interest the existence of gender inequalities in the effect of social networks on income. The fact that no significant linkage is found for women-headed households could be attributable to gender differences in the structure and composition of the social networks. In [48], the authors found that women's social networks consist of mostly relatives and female neighbors, while the social networks of men are mainly formed by job- and business-related acquaintances. This could also explain our results in the case of rural Mexico.

Regarding the significant association of holding a credit card with the income-to-poverty ratio, this result is in accordance with previous studies for other countries [16, 20]. There are two potential interpretations of this fact. On the one hand, there might be a causal impact of having access to financial services on income through the application of economic resources from credits directed at entrepreneurial investment, smooth consumption, protection against income or price shocks, and resource allocation. On the other hand, having an insufficient income is at the same time a constraint for accessing financial services.

About the type of household, one-person households show the greatest income-to-poverty ratio compared to nuclear, extended, and other family structures. Our results could suggest that the larger the number of dependents per family, the lower the income, which matches well with [1] and [29].

There is also a clear indication that the multiple dimensions of poverty are interlinked in rural communities. The significant correlation between access to food and income poverty may reveal a two-way relationship. First, consuming a nutritious diet requires allocating enough money for buying adequate food in quality and quantity, which can be a challenge for households struggling with income-related adversities. Second, consumption of nutritious food helps to maintain good health, and in turn, it improves the ability of individuals to join the labor force, obtain a better job or achieve higher productivity.

Not holding the mandatory educational level (education lag) is another aspect of multidimensional poverty associated with income for rural poor and extremely poor families. Although more detailed statistical information is required to inquire into the reasons for this result, as stated in [49], a potential explanation could be that completing compulsory education increases income because this education level is associated with a fertility rate reduction and a health status improvement through an increase in contraceptive use and a delay of the age of marriage (or age of first pregnancy), which in turn is linked to a higher productivity.

Having a house with access to basic services is positively associated with income in rural households, and this is equally affecting both women- and men-headed households. These results share similarities with the findings from [16]. Housing represents a major charge on income. Given that the poor and extremely poor families have an income that is not enough to get an affordable and decent house, the poorest of the poor could be being forced to live in places with more marginalized circumstances such as a lack of access to services. Simultaneously, these conditions could be affecting their health, education, access to work, and productivity.

Our results also support the existence of a positive relationship between income and access to the social protection system. This association could indicate that to have access to social

security services, the head of the family must have a formal working contract, and formal jobs in Mexico are usually better paid than non-formal jobs [50]. It is worth noting that the results also point to gender inequalities against women-headed households, which could be corroborating the fact that in Mexico women face greater difficulties in entering the formal job market than men [51].

A commonsense result is achieved when analyzing the association of education level with the income-to-poverty ratio. Consistent results highlight that families whose head completed at least university have a higher income. Our results also indicate that for families having a woman head with a medium level of education, education has an age-varying effect described by an inverted U-shaped curve. For men-headed families, the relationship is described by a downward-sloping line. The form of these shapes reflects how the income evolves heterogeneously over the life cycle, maybe due to differences in working productivity.

As expected, our results confirm significant differences according to the marital status of the head. However, unlike other research in this area indicating that lower-income levels are observed in households that experienced a dissolution [16, 52], we find a greater income-to-poverty ratio for households headed by a separated man. Remarkably, this effect is not shown for families with a divorced head. A possible explanation for this may be that when separating from their partners, men have an increase in their household income as a result of a decrease in the number of dependents and because in contrast to divorced heads, there is no legal judgment dissolving the union between the male head and his ex-partner, and therefore the legal responsibilities for them, such as financial support for dependents, are not delineated.

Another interesting result on gender issues in this paper is weekly time spent by the head on housework. This covariate is found to be significant for poor households headed by a man. Our results substantiate previous findings for other countries [18, 53]. It is logical to analyze this association as a trade-off between the time devoted by the head to paid and unpaid work and as a trade-off of time among the family members. Our results show that when all the work is done by the rest of the household members, the income-to-poverty ratio is low, maybe because the head is the only member with time available to engage in a paid activity. As the head expends more time on housework, income raises possibly because more members spend less time on housework but more time on the market work. However, after a maximum point (between 10 and 15 hours per week), the head cannot increase his housework time without decreasing the income-to-poverty ratio of the family.

Regarding the characteristics of the community of residence, findings indicate that households in more unequal rural communities tend to exhibit lower income levels. Similarly, greater income-to-poverty ratios are observed in municipalities with the best levels of human development. These correlations could be revealing that the favorable living conditions of the communities (income equality and human development) have a positive effect on the income of the poor and extremely poor households [54]. At the same time, these associations can also reflect the residential decisions of the households: families with enough money decide to move to municipalities with better living conditions, while those with the worst income levels remain in the communities with the worst living conditions.

Regarding the correlation of women's household headship in the municipality of residence with the income-to-poverty ratio, we find a particularly intriguing result. It is well known that in patriarchal societies, assigning a woman as a head is an unusual situation frequently linked to lone-parent households or childbearing outside the marriage, so it can be expected that in municipalities with these adverse circumstances against women, families headed by a woman have on average a lower income [55]. In contrast to this general assumption, we find that after controlling for the individual-, community- and regional-level characteristics, women's headship in the community and income-to-poverty ratio have a positive relationship in extremely

poor households headed by women. Apart from residential decisions of women-headed families to live in municipalities where more women are acknowledged as heads of the household, it could also indicate that in these communities, women have greater well-being and empowerment that impacts on income. Alternatively, it can also suggest that they are receiving remittances from the partner.

Moreover, our results indicate that as the share of women involved in economic activity of the rural community rises, higher income levels are observed not only for women-headed households but also for men-headed households in extreme poverty. This could suggest that the inclusion of women in the economically active population helps address labor market imbalances in rural communities, expands the working-age population, or contributes to boosting the human capital, which impacts household income. It could also be that these families decide to reside in communities where the women have greater employment opportunities.

The examination carried out in this paper also reveals that the quality of the public provision of goods and services in the region of residence is positively associated with the income-to-poverty ratio. On the one hand, it can indicate that families being the poorest of the poor tend to have a residence in regions with lower quality of public services (which are likely to have a lower cost of living), while those who have an income enough to afford it reside in a region with better provision of public services. On the other hand, it may suggest that the provision of public services impacts the income of the households via an improvement in their quality of life [27].

Finally, another interesting result on gender issues at the regional level is the linkage between gender-based violence against women and the income-to-poverty ratio. As described in the previous section, higher income levels are associated in rural areas with increases in the share of women victims of gender-based violence in the public sphere. It is particularly important to observe that this correlation is uniquely relevant for men-headed families. Given that it is well known that gender-based violence is the result of the exercise of men's power over women and girls, a probable explanation is that male heads living in a family with an income that can afford to move to another state, seek to live in communities where they can exercise their domination. A more detailed analysis of this matter is outside the scope of this paper and therefore left for further research.

## Conclusions

This paper aims to identify a set of relevant variables associated with the income-to-poverty ratio in rural Mexican households. We emphasize finding the extent to which the effect of the significant factors differs between women- and men-headed households and how these gendered effects vary according to the depth of poverty experienced by the families.

To achieve this goal, we construct a cross-sectional dataset containing information on 4,434 women-headed and 14,877 men-headed households to which we incorporate 45 variables at the individual/household, community, and regional levels, from nine different data sources. This dataset is used to estimate four additive quantile models, which allow us to compute specific parameters for different quantiles of the income-to-poverty ratio distribution. In particular, two models are applied to data on households headed by a woman and are estimated for the quantiles corresponding to the poor and extremely poor families. Similarly, the other two models correspond to poor and extremely poor man-headed families.

Based on the association of the considered variables with the income-to-poverty ratio, the results presented herein allow us to distinguish two different main types of effects. First, we identify a subset of variables whose significance is consistent for poor and extremely poor

families, but their effect on income is not statistically different between women- and men-headed households. These variables are credit card ownership, access to basic housing services, education level, and satisfaction with public services.

Second, our results also identify significant differences between women- and men-headed families concerning the effects of several variables on the income-to-poverty ratio for poor and extremely poor households. Variables belonging to this group are social networks, access to social security, Gini index, human development, and gender-based violence against women in the public sphere. More importantly, for social networks, access to social security, and gender-based violence against women in the public sphere, the uneven effect between the sexes grows as family income goes from extreme poverty up to the poverty level.

Broadly speaking, these results have key implications on the study of income poverty in rural Mexico through a gendered lens. By controlling by a large set of factors at the individual/household, community and region levels, our results help us to underscore the circumstances in which women- and men-headed households face particular disadvantages. In this regard, we detect some households, traditionally overlooked, that may experience even worse poverty levels. These are, among others, households headed by an older man, families having a younger or older woman head with a medium level of education, men-headed households lacking social networks, and extended households headed by a woman. Differently, is it worth noticing that having a highly educated woman as head of the household seems to be related with lower severity of poverty. This result emphasizes the importance of women's education as a mean of fighting poverty in rural areas.

Although the response variable used in this paper, income-to-poverty ratio, enable us to capture how the effect of different predictors varies with the severity of poverty, further research should also consider other poverty indicators, such as those related to multidimensional poverty, or even analyze other distributional parameters of the response, including the scale and shape. Limitations of our research are related to the inherent characteristics of cross-sectional data. First, results only apply to rural Mexico in the period of reference. Moreover, income poverty is both a cause and an effect of many covariates included in this paper, being difficult to determine to what extend the selected covariates are causing an increase or decrease in the ratio of income-to-poverty. In any case, our methodological approach and findings can be used as a reference for further research on the matter.

## Supporting information

**S1 Table. Summary statistics of the income-to-poverty ratio by sex of the head.** Note: Summary statistics are calculated for each dataset considering the full distribution of the income-to-poverty ratio corresponding to women- or men-headed households.
(XLSX)

**S2 Table. Summary statistics of the categorical covariates by sex of the head.**
(XLSX)

**S3 Table. Summary statistics of the continuous covariates at the individual level by sex of the head.**
(XLSX)

**S4 Table. Summary statistics of the continuous covariates at the community level by sex of the head.** Note: Summary statistics at the community level are calculated considering 513 municipalities both for women- and men-headed households.
(XLSX)

**S5 Table. Summary statistics of the continuous covariates at the regional level by sex of the head.** Note: Summary statistics at the regional level are calculated considering 32 states both for women- and men-headed households.
(XLSX)

**S6 Table. Summary of estimated coefficients in terms of standard deviations for covariates with stable significant effects and their 95% confidence intervals (CI).**
(XLSX)

## Author Contributions

**Conceptualization:** Juan Armando Torres Munguía, Inmaculada Martínez-Zarzoso.

**Data curation:** Juan Armando Torres Munguía.

**Formal analysis:** Juan Armando Torres Munguía, Inmaculada Martínez-Zarzoso.

**Investigation:** Juan Armando Torres Munguía.

**Methodology:** Juan Armando Torres Munguía, Inmaculada Martínez-Zarzoso.

**Software:** Juan Armando Torres Munguía.

**Supervision:** Inmaculada Martínez-Zarzoso.

**Visualization:** Juan Armando Torres Munguía.

**Writing – original draft:** Juan Armando Torres Munguía, Inmaculada Martínez-Zarzoso.

**Writing – review & editing:** Juan Armando Torres Munguía, Inmaculada Martínez-Zarzoso.

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
