## [Decision Letter · Decision Letter 0]

1 Apr 2021

PONE-D-21-04773

Examining gender inequalities in factors associated with income poverty in Mexican rural households by a boosting additive quantile model with stability selection

PLOS ONE

Dear Dr. Torres Munguía,

Thank you for submitting your manuscript to PLOS ONE. After careful consideration, we feel that it has merit but does not fully meet PLOS ONE’s publication criteria as it currently stands. Therefore, we invite you to submit a revised version of the manuscript that addresses the points raised during the review process.

We look forward to receiving your revised manuscript.

Kind regards,

Carlos Alberto Zúniga-González, Ph.D

Academic Editor

PLOS ONE

Journal Requirements:

2. Please change "female” or "male" to "woman” or "man" as appropriate, when used as a noun (see for instance https://apastyle.apa.org/style-grammar-guidelines/bias-free-language/gender).

3. We note you have included a table to which you do not refer in the text of your manuscript. Please ensure that you refer to Table 5 in your text; if accepted, production will need this reference to link the reader to the Table.

Additional Editor Comments (if provided):

Dear authors, I believe that the improvements indicated by the reviewers should be made, mainly because of the type of manuscript where a way of considering various variables is combined.

Reviewers' comments:

Reviewer's Responses to Questions

**Comments to the Author**

1. Is the manuscript technically sound, and do the data support the conclusions?

Reviewer #1: Partly

Reviewer #2: Partly

2. Has the statistical analysis been performed appropriately and rigorously? 

Reviewer #1: I Don't Know

Reviewer #2: Yes

3. Have the authors made all data underlying the findings in their manuscript fully available?

Reviewer #1: Yes

Reviewer #2: Yes

4. Is the manuscript presented in an intelligible fashion and written in standard English?

Reviewer #1: No

Reviewer #2: Yes

5. Review Comments to the Author

Reviewer #1: Summary: This is an ambitious paper that presents a complex analytical approach to investigate associations between a large number of individual and higher-level variables and poverty. While interesting, the paper is written in a highly technical manner at times that is somewhat difficult to understand without having a specific understanding of the statistical methods used. Further, the authors use a fair amount of statistical jargon without clearly explaining how to understand specific terms vis a vis the analysis or understand them in the context of results. Last, there many grammatical errors throughout that make the paper difficult to understand in places. I’ve tried to highlight a few below, but there many and the manuscript needs a very close read by a native English speaker.

Page 1: Grammar – “Broadly speaking, there exist [exists] a consensus on the fact that old-age...”

Page 2: Grammar – “marginalization and social deprivations are associated to [with] larger [higher]

poverty levels”

Table 2: Several of the variables are not clearly defined. Given that much of the interpretation of the results are clearly dependent on how the variables are specified, it is important that all the variables in the model are described in detail and how any categorizations were determined. For example, please more clearly define how social networks is categorized into a single measure of three categories, how “educational lag” is defined. Similarly, “Social marginalization level in 2015.”

Results: The authors combine much of the discussion that is typically included in a discussion section into the results section. As a result, it is sometimes difficult to differentiate between what is the author’s conjecture or interpretation of the results, versus what the data suggest. The authors may want to consider including a discussion section to allow for this discussion of the results.

Throughout the results section, the authors use much technical language that makes it difficult for a more general interest readership. Suggestion for the authors to define more technical terms, such as “offset” (page 17) or explain how to interpret it and why it matters.

Page 12: “This association is statistically larger for poor families than for extremely poor households.” What does this mean? Did the authors do any formal statistical tests or contrasts to explore whether these coefficients are indeed statistically different?

Page 12: “The fact that no significant correlation is found for female-headed households could be

attributable to differences in socialization patterns between sexes in rural communities, which in general tend to be more traditional in terms of gender norms and roles.” The point being made here is unclear. Why would traditional gender norms and roles influence perceptions of social networks? Authors could explain this more. It would further be helpful to understand how the variable was fully defined.

On page 17, the authors mention that the reader should review Table 2 for definitions of key variables. However, this is somewhat circular, as per my previous comment, several of the variables are poorly defined in Table 2. For example, “Share of female victims of violence in the workplace.” How was this variable measured? What constitutes violence in the workplace? Was this ascertained then aggregated from self report or some other means

The paper would be improved by a clearer organization of the conclusions section. As written, it is somewhat difficult to follow, and it is unclear as to whether the importance of this paper rests primarily in the use of this specific analytical approach or if there are other important substantive results from the paper.

Reviewer #2: This article investigates poverty using, author say, a novel approach with statistical techniques such as quantile regression and techniques to deal with high dimensionality. I think the paper should be carefully revised to make sure that readers are not lost while reading it. Also it should be made clear whether the paper is about poverty or about a technique and frame accordingly. Major revisions are expected. I think authors should make the effort as I think this work has potential and I am eager to read a second version of this paper.

Major Comments

I have had difficulties to understand the dependent variable. Even though I hold a master in statistics, I think authors should not expect reader to be familiar with quantile regression as one can be with linear or binary regression models. Therefore, I think it should be carefully brought to a reader. What a quantile regression does is that for each subset of the continuous dependent variable it computes a parameter. The subsets are defined by quantiles. It seems from Table 1 that there are three subsets defined at the following quantiles: 0.149 and 0.411. I do not think Table 1 is very useful, it is actually misleading as it gives the impression that the dependent variable is a categorical variable with two or three categories. I do not see the usefulness of the ‘Type of community’.

Following on the previous paragraph I understand with the results that the authors have had only two quantile regressions (times 2 for gender), which I do not understand how can it be so, each X should have 3 parameters one for below the first threshold; one for below the second and above the first and one for above the second. It does not seem to be the case. Authors should explain this more clearly in the paper. This and the above paragraph can be answered together.

I think the results section should be separated in three: when there are no differences in point estimate of X between the parameter for extremely poor and the one for poor. Then another part that should be emphasized because it is the main added value of the paper is that when parameters are different (when one is significant and one is not) across the two regression. Finally differences in parameter for female head and male heads are discussed/presented in a third section. I authors think differently they should explain carefully why not.

I understand that the modelling was hard work but however hard it is, this journal is not a journal for methods in research but should show results that are relevant in public health/social science. Therefore the main contribution cannot be the hard work regarding the methods. So the paper, the whole paper, should be framed so that one or two main results regarding poverty literature should be put ahead as major contributions. The methods would obviously be in the background because it is the method that would allow this major contribution to be made. What is the key results in the poverty literature that the method allow the authors to claim as major? => I would say it should be a difference in parameter estimates between the below the threshold of extremely poor and poor therefore showing that the technique brings something. I the aim is to be have a statistical paper, then it should why this technique is new. I am not sure this is the goal.

Minor Comments/Reformulation

Page 4: “Similarly, a person is considered to be poor if his/her income” => I would remove ‘similarly’ as it is misleading.

Page 4: “Based on these two lines, the dependent variable introduced” => “Based on these two thresholds, the dependent variable introduced” change lines to threshold.

Page 4 “associated covariates are chosen from previous research on the matter and include a wide variety of variables at the individual, household, community, and regional levels” => “associated covariates are chosen from previous research on the matter and include characteristics at the individual, household, community, and regional levels”, shorten the variables by replacing the ‘wide variety …’ by ‘characteristics’.

Page 8, ‘four quantile regression models are estimated.’ => maybe state in one small sentence which are those.

Page 12: “which in general tend to be more traditional in terms of gender norms and roles.” => a citation should be added for this or not try to discuss this point at this stage.

6. PLOS authors have the option to publish the peer review history of their article (what does this mean?). If published, this will include your full peer review and any attached files.

Reviewer #1: No

Reviewer #2: **Yes: **Simon Combes

---

## [Author Response · Author response to Decision Letter 0]

6 Jul 2021

Department of Economics

Georg-August-Universität Göttingen

Platz der Göttinger Sieben 3

37073 Göttingen

Germany

July 5, 2021

Dear Carlos Alberto Zúniga-González, 

We would like to thank you very much for inviting us to revise and resubmit our manuscript “Examining gender inequalities in factors associated with income poverty in Mexican rural households”. We appreciate the time and effort that you and the reviewers have dedicated to providing insightful comments and suggestions that have helped us to improve the quality and readability of the manuscript. 

Following your advice, we have clarified the main research questions and framed the contributions of the paper to the existent literature on poverty as well as modified the wording to make it less technical and more comprehensible to readers. In what follows, we provide detailed response to each comment, indicating the specific changes made in the manuscript to address them and the location where those are placed in the revised manuscript. In order to facilitate the follow-up of these changes, we have included line numbers in the revised manuscript with track changes.

We hope that the revised version fulfils your expectations and that the new structure and organization of the paper is now clearer.

Sincerely,

The authors

Comments from the Academic Editor

Comment 1: 

Response: 

 Thank you. We revised again the documents following the sample files for the title, author list, and affiliations page; and the manuscript body to ensure that our submission meets the formatting requirements of the journal. 

As part of these changes, we shortened the title of the manuscript to “Examining gender inequalities in factors associated with income poverty in Mexican rural households” (before “Examining gender inequalities in factors associated with income poverty in Mexican rural households by a boosting additive quantile model with stability selection”) aiming to make it more specific, descriptive, concise, and comprehensible to readers outside the field, as required in the guidelines for the title, and to highlight that the main goal of the manuscript rests primarily in study of poverty and not in this specific analytical approach. 

Comment 2: 

 Please change "female” or "male" to "woman” or "man" as appropriate, when used as a noun.

Response: 

 Thank you for pointing this out. In this revised version we changed all the times we have used "female” or "male" as a noun to "woman” or "man". As suggested, we followed the general guidelines for talking about gender with inclusivity and respect covered in Section 5.5 of the APA Publication Manual, Seventh Edition.

Comment 3: 

 We note you have included a table to which you do not refer in the text of your manuscript. Please ensure that you refer to Table 5 in your text; if accepted, production will need this reference to link the reader to the Table.

Response: 

 Thank you. We have ensured that all Tables and Figures included in the manuscript have a reference in the text. 

Comments from Reviewer 1

Comment 1: 

 Summary: This is an ambitious paper that presents a complex analytical approach to investigate associations between a large number of individual and higher-level variables and poverty. While interesting, the paper is written in a highly technical manner at times that is somewhat difficult to understand without having a specific understanding of the statistical methods used. Further, the authors use a fair amount of statistical jargon without clearly explaining how to understand specific terms vis a vis the analysis or understand them in the context of results.

Response: 

 Thank you, we appreciate the comments. We have used a clearer and less technical wording in the revised paper and added explanations for the methods used. We have reworded the Materials and Methods section, particularly the part regarding the Model specification (pages 16-20 of the revised manuscript with track changes). We avoided the use of statistical jargon and specific terms. Instead, we included in this part a description of the major advantages offered by the modelling design in the context of poverty research (see lines 224-242); and, a non-technical description of the application of the boosting algorithm, stability selection and the confidence intervals (lines 288-329).

Comment 2: 

 Last, there many grammatical errors throughout that make the paper difficult to understand in places. I’ve tried to highlight a few below, but there many and the manuscript needs a very close read by a native English speaker.

Response: 

 Thank you. We have corrected the grammatical errors pointed out. In addition, we sent our manuscript to revision by a native English speaker professional in academic writing.

Comment 3: 

 Page 1: Grammar – “Broadly speaking, there exist [exists] a consensus on the fact that old-age...”

Response: 

 Change made (see line 61)

Comment 4: 

 Page 2: Grammar – “marginalization and social deprivations are associated to [with] larger [higher] poverty levels”

Response: 

 Change made (see line 63)

Comment 5: 

 Table 2: Several of the variables are not clearly defined. Given that much of the interpretation of the results are clearly dependent on how the variables are specified, it is important that all the variables in the model are described in detail and how any categorizations were determined. For example, please more clearly define how social networks is categorized into a single measure of three categories, how “educational lag” is defined. Similarly, “Social marginalization level in 2015.”

Response: 

 We totally agree. The table with the definition of variables has been restructured in the revised manuscript. As shown in pages 11-16, now we have included 5 columns: Level (individual/household, community or regional), Variable (and type of variable), Definition, Relationship types, and Source. Particularly, in the column Definition we included an explanation of the concept, a brief description of the methodology, and the definition of the categories for categorical variables.

Comment 6: 

 Results: The authors combine much of the discussion that is typically included in a discussion section into the results section. As a result, it is sometimes difficult to differentiate between what is the author’s conjecture or interpretation of the results, versus what the data suggest. The authors may want to consider including a discussion section to allow for this discussion of the results.

Response: 

 We fully agree and thanks you very much for this suggestion. To make this point we added a discussion section (pages 33-38) in which we have included our interpretation of the results. The results section (pages 20-33) only includes comments on the estimates, i.e. what the data suggest.

Comment 7: 

 Throughout the results section, the authors use much technical language that makes it difficult for a more general interest readership. Suggestion for the authors to define more technical terms, such as “offset” (page 17) or explain how to interpret it and why it matters.

Response: 

 Thank you, we agree with the comment. In the revised version, we have reworded the results section, avoiding the use of statistical / technical terms that were not contributing to the understanding or interpretation of the results. Part of the changes made is that we have included all the results in a single table, instead of having three different tables. To facilitate the lecture of Table 2 (and also following the suggestion from Reviewer 2 about highlighting the coefficients statistically different between women- and men-headed households, and among poverty levels) we only included the results with a significant effect, and we indicate in bold letters when the covariate effect on the response variable varies with household income level, and in grey __ we indicate that the covariate effects on the response variable varies according to the sex of the household head. We also included more figures to visualize the comparison between estimated effects. In this way, by focusing exclusively on the significant effects, adding shades, bold letters, and more figures, it is easier to appreciate the differences in the estimated coefficients without using technical terms. 

See pages 20-33 corresponding to results, and pages 33-38 for the discussion section. Table 2, with all the results is found in pages 11-16.

Comment 8: 

 Page 12: “This association is statistically larger for poor families than for extremely poor households.” What does this mean? Did the authors do any formal statistical tests or contrasts to explore whether these coefficients are indeed statistically different?

Response: 

 We compare the coefficients by estimating their 95% confidence intervals. To compute the confidence intervals, 1000 random samples from the empirical distribution of the data are used. We include the confidence intervals for significant effects in Table 2, which includes a summary of the estimated coefficients for variables with stable significant effects and their 95% confidence intervals (CI). We added up to 3 decimal points to the estimations to avoid inaccuracies when comparing and referring to statistical differences. It helped us, for instance, to make more visible that the parameter estimations of the effect of covariate credit card on the response for women-headed households living in extreme poverty -0.114 [-0.165, -0.05] and poverty -0.224 [-0.29, -0.163] are not statistically different. Moreover, when commenting the results, we added the estimated confidence intervals for reference. See pages 20-33 corresponding to results.

Comment 9: 

 Page 12: “The fact that no significant correlation is found for female-headed households could be attributable to differences in socialization patterns between sexes in rural communities, which in general tend to be more traditional in terms of gender norms and roles.” The point being made here is unclear. Why would traditional gender norms and roles influence perceptions of social networks? Authors could explain this more. It would further be helpful to understand how the variable was fully defined.

Response: 

 In order to address your comment, we have proceeded as follows. First, as mentioned before, we added a discussion section in which we have included our interpretation of the results. Second, to make interpretations clearer we included references to previous studies in the discussion. Third, regarding the definition of variables, we have reorganized the table with the name and definition for each factor considered in the full model (see pages 11-16). 

For the particular case of social networks, the definition can be found in in Table 1 (page 11 of the revised manuscript). This table also includes a reference to the official criteria for creating the three categories. Lines 527-533 include the description of the results, whereas lines 704-714 include our interpretation of the results, where we also provide references to previous studies that pointed to the existence of a different structure and composition of the social networks of women and men (lines 706 and 711).

Comment 10: 

 On page 17, the authors mention that the reader should review Table 2 for definitions of key variables. However, this is somewhat circular, as per my previous comment, several of the variables are poorly defined in Table 2. For example, “Share of female victims of violence in the workplace.” How was this variable measured? What constitutes violence in the workplace? Was this ascertained then aggregated from self report or some other means

Response: 

 As mentioned before, we have reformulated the table with the list of variables and included on it a definition for each one (see pages 11-16). Specifically, the variable “Share of female victims of violence in the workplace” is renamed as “Gender-based violence against women and girls in the workplace”. In the second column, the definition is “% of the 2016 women’s population aged ≥ 15 years who were victims of psychological, physical, and/or sexual gender-based violence in the workplace between October 2015 and October 2016 in the region of household residence. Expressed in decimal form.” We also included the source, the National Survey on the Dynamics of Household Relationships, and the link to the data for more details in the Bibliography. 

Comment 11: 

 The paper would be improved by a clearer organization of the conclusions section. As written, it is somewhat difficult to follow, and it is unclear as to whether the importance of this paper rests primarily in the use of this specific analytical approach or if there are other important substantive results from the paper.

Response: 

 We fully agree and thank you for this comment, which is taken on board. We have reorganized the conclusions section as follows. First, we highlight that our paper main goal is to contribute to the studies on poverty, not the use of a specific analytical approach (lines 830-834). Second, we refer to the dataset used and the models estimated (lines 835-843). Third, we emphasized more the findings on poverty by differentiating between factors that affect equally women and men headed households and those that have a differential effect (lines 845-898). Finally, we end indicating some limitations and suggestions for further research (lines 899-905).

Comments from Reviewer 2

Comment 1: 

 This article investigates poverty using, author say, a novel approach with statistical techniques such as quantile regression and techniques to deal with high dimensionality. I think the paper should be carefully revised to make sure that readers are not lost while reading it. Also it should be made clear whether the paper is about poverty or about a technique and frame accordingly. Major revisions are expected. I think authors should make the effort as I think this work has potential and I am eager to read a second version of this paper.

Response: 

 Thank you for the constructive review. We agree with the comment and have made the following changes: First, we have changed the title to highlight that our paper main goal is to contribute to the studies on poverty, and not the application of the specific technique. Second, we have reworded and clarified the document, highlighting the findings on poverty. Third, we have restructured the model specification to avoid using statistical jargon. Fourth, we have restructured the results section and added a discussion section. Finally, the conclusions have been reframed to emphasize that the paper is about poverty.

Comment 2: 

 I have had difficulties to understand the dependent variable. Even though I hold a master in statistics, I think authors should not expect reader to be familiar with quantile regression as one can be with linear or binary regression models. Therefore, I think it should be carefully brought to a reader. What a quantile regression does is that for each subset of the continuous dependent variable it computes a parameter.

Response: 

 Thank you for pointing this out. We agree that our submission was myopic in terms of clarifying the quantile regression applied to poverty analysis. In order to make this point, we have rewritten the model specification section (lines 224-330). In lines 225-242 we clarified how the additive structure, and the quantile approach proceed. Particularly for the quantile approach, in lines 108-119 of the introduction we added a general description of the use of the quantile approach in the paper, and in lines 238-242 we provided further explanation before expressing formally the model. 

Comment 3: 

 The subsets are defined by quantiles. It seems from Table 1 that there are three subsets defined at the following quantiles: 0.149 and 0.411. I do not think Table 1 is very useful, it is actually misleading as it gives the impression that the dependent variable is a categorical variable with two or three categories. I do not see the usefulness of the ‘Type of community’.

Response: 

 We agree, accordingly, we have deleted Table 1 (page 7) and described in the text the information on the quantiles of the dependent variable (see lines 174-194). 

Comment 4: 

 Following on the previous paragraph I understand with the results that the authors have had only two quantile regressions (times 2 for gender), which I do not understand how can it be so, each X should have 3 parameters one for below the first threshold; one for below the second and above the first and one for above the second. It does not seem to be the case. Authors should explain this more clearly in the paper. This and the above paragraph can be answered together.

Response: 

 As explained in the last two comments, we reformulated the text about the description of the income-to-poverty variable (see lines 174-194). In this paper, we exclusively focus on two quantiles: one corresponding to people living in extreme poverty (below the extreme poverty threshold) and the other refer to people living in poverty (below the poverty threshold). 

Comment 5: 

 I think the results section should be separated in three: when there are no differences in point estimate of X between the parameter for extremely poor and the one for poor. Then another part that should be emphasized because it is the main added value of the paper is that when parameters are different (when one is significant and one is not) across the two regression. Finally differences in parameter for female head and male heads are discussed/presented in a third section. I authors think differently they should explain carefully why not.

Response: 

 We have added the suggested structure to the manuscript. The results section was fully restructured. We reframed this, including the main table of results (Table 2 of the revised manuscript with track changes), to highlight the differences between sex of the household-head, and between poor and extremely poor households. We have proceeded as follows. First, we refer to the variables for which the estimated coefficients for women-headed and men-headed families are not statistically different (lines 376-443), and within this section we comment those whose effect is statistically different between poverty levels. Finally, in lines 446-696 we present the results in which the estimated parameters for women-headed and men-headed households are statistically different, and among them those with a statistically different parameter effect between poverty levels. 

Comment 6: 

 I understand that the modelling was hard work but however hard it is, this journal is not a journal for methods in research but should show results that are relevant in public health/social science. Therefore the main contribution cannot be the hard work regarding the methods. So the paper, the whole paper, should be framed so that one or two main results regarding poverty literature should be put ahead as major contributions. The methods would obviously be in the background because it is the method that would allow this major contribution to be made. What is the key results in the poverty literature that the method allow the authors to claim as major? => I would say it should be a difference in parameter estimates between the below the threshold of extremely poor and poor therefore showing that the technique brings something. I the aim is to be have a statistical paper, then it should why this technique is new. I am not sure this is the goal.

Response: 

 Thank you for pointing this out. We have reframed the manuscript to emphasize that our goal is to contribute to the studies on poverty. A suggested, we highlight two major contributions. First, the identification of a subset of significant factors whose effect is independent of the head’s sex and is relevant for poor and extremely poor families. Second, the identification of a subset of significant factors with an uneven effect on income according to the sex of the head that is observed both in the poor and extremely poor households. 

Comment 7: 

 Page 4: “Similarly, a person is considered to be poor if his/her income” => I would remove ‘similarly’ as it is misleading.

Response: 

 Change made (line 185).

Comment 8: 

 Page 4: “Based on these two lines, the dependent variable introduced” => “Based on these two thresholds, the dependent variable introduced” change lines to threshold.

Response: 

 Since we deleted Table 1 as suggested in Comment 3, this paragraph has also been deleted (lines 195-196). The information related is reworded in the lines 174-194.

Comment 9: 

 Page 4 “associated covariates are chosen from previous research on the matter and include a wide variety of variables at the individual, household, community, and regional levels” => “associated covariates are chosen from previous research on the matter and include characteristics at the individual, household, community, and regional levels”, shorten the variables by replacing the ‘wide variety …’ by ‘characteristics’.

Response: Change made (line 200 and 2001).

Comment 10: 

 Page 8, ‘four quantile regression models are estimated.’ => maybe state in one small sentence which are those.

Response: 

 We reworded the section on the model specification. The suggestion on specifying which models are the ones estimated has been incorporated on lines 243-248.

Comment 11: 

 Page 12: “which in general tend to be more traditional in terms of gender norms and roles.” => a citation should be added for this or not try to discuss this point at this stage.

Response: 

 We have included a discussion section in which we link our findings to previous research. In particular, the discussion on social networks can be found on lines 704-714. The specific citation on this regard is found on line 711. We have also reworded this to clarify our point.

---

## [Decision Letter · Decision Letter 1]

17 Aug 2021

PONE-D-21-04773R1

Examining gender inequalities in factors associated with income poverty in Mexican rural households

PLOS ONE

Dear Dr. Juan Armando Torres Munguía

Thank you for submitting your manuscript to PLOS ONE. After careful consideration, we feel that it has merit but does not fully meet PLOS ONE’s publication criteria as it currently stands. Therefore, we invite you to submit a revised version of the manuscript that addresses the points raised during the review process.

We look forward to receiving your revised manuscript.

Kind regards,

Carlos Alberto Zúniga-González, Ph.D

Academic Editor

PLOS ONE

Journal Requirements:

Additional Editor Comments (if provided):

Dear author, I have checked that the comments of the reviewers have been incorporated, however there are minor revisions to perform indicated by the reviewer. I suggest you review this reference that I believe will help you improve the approach of the methodology used referred to by the reviewer Zuniga González, C. A., & Jaramillo Villanueva, J. L. (2012). Wages and Employs for Non-Farm Agricultural Activities: One Livelihood Strategy in Nicaragua. Global Journal of Management And Business Research, [S.l.], v. 12, n. 15, sep. 2012. ISSN 2249-4588. Available at: <https: 784="" article="" gjmbr="" index.php="" journalofbusiness.org="" view="">. Date accessed: 15 aug. 2021.

Reviewers' comments:

Reviewer's Responses to Questions</https:>

**Comments to the Author**

1. If the authors have adequately addressed your comments raised in a previous round of review and you feel that this manuscript is now acceptable for publication, you may indicate that here to bypass the “Comments to the Author” section, enter your conflict of interest statement in the “Confidential to Editor” section, and submit your "Accept" recommendation.

Reviewer #2: All comments have been addressed

2. Is the manuscript technically sound, and do the data support the conclusions?

Reviewer #2: Yes

3. Has the statistical analysis been performed appropriately and rigorously? 

Reviewer #2: Yes

4. Have the authors made all data underlying the findings in their manuscript fully available?

Reviewer #2: Yes

5. Is the manuscript presented in an intelligible fashion and written in standard English?

Reviewer #2: Yes

6. Review Comments to the Author

Reviewer #2: Comments

Very minor

Line 320 page 17: "About variable age, it has a" => "Age has a" ?

Minor Comments

Page 2 line 36-39: “On the other hand,  most of the research is exclusively based on mean regression models analyzing the population’s average income or the expected probability of being poor, disregarding the potential inequalities across income levels. These inequalities justify the need to focus on the poorest.” => proposal : “On the other hand,  most of the research is exclusively based on mean regression models analyzing the population’s average income or the expected probability of being poor, disregarding specific effects in poorer income levels. We propose to fill this gap by focusing on the poorest” <= I think this is what the paper does and the "potential inequalities" is a bit vague. I understand that the author may have try to go slowly in the introduction and to not give away too much too early, but the word "inequality" may mislead a reader as "inequality" is nearly quite a domain, I do not think the paper is actually in this domain it is in the literature on poverty (which is linked to inequality, do not get me wrong). If the paper is on inequalities I would expect that the paper looks at rich and poor while it specifically look at poor. I can change my mind if the authors bring some literature explaining carefully the link between poverty and inequalities (I am not saying it is not linked just that it is not that straightforward).

Table 2, some cells are gray with nothing in it, is that on purpose? (eg. second row first column)how can be estimated the difference between men and women if the parameter is not estimated?

Line 254 on page 15: "For extremely poor families, the coefficient for households without access to credit cards and with a woman as the head is -0.114" => can you tell us what is this parameter? it increase the income to poverty ratio by 11%? Maybe tell the reader that people with credit card are less poor? is that what it means? If I am mistaken then this comment should be taken as evidence that the paragraph is not very clear.

Complementary to my previous comment, I think the authors should translate the parameters into meaningful effects, are parameters percentages? increases in standard deviation? at the end of the result section, page 21, line 418, it seems that the effect is an increase in the ratio, is that the case ? Then even this would not be very informative, could it be that the parameter shows the effect as the number of standard deviation of the dependent variable?

With OLS it would necessitate to divide the dependent by its own standard deviation and then the parameter would be for a binarry variable like sex, being a female (if male the ref) increase by Beta (eg Beta being positive) standard deviation the dependent variable. If beta is above one then it increase by more than a standard variation , if below it increases by less than one standard variation. But I may not understand well enough the quantile regression

line 455, page 22 When citing the world bank report on education, a little bit of the explanation of why higher education reduces fertility would be good, so could you add someting like education reduces fertility because it increases contraception and age at marriage?

7. PLOS authors have the option to publish the peer review history of their article (what does this mean?). If published, this will include your full peer review and any attached files.

Reviewer #2: **Yes: **Simon (Jean-Baptiste) Combes

---

## [Author Response · Author response to Decision Letter 1]

13 Oct 2021

Department of Economics

Georg-August-Universität Göttingen

Platz der Göttinger Sieben 3

37073 Göttingen

Germany

October 8, 2021

Dear Dr. Carlos Alberto Zúniga-González, 

Thank you very much for giving us the opportunity to resubmit our revised manuscript entitled “Examining gender inequalities in factors associated with income poverty in Mexican rural households”. We are grateful to you and the reviewers for the time and effort dedicated to our document, for the improvement, precision, and quality that you brought to the manuscript.

We have been able to incorporate changes to reflect all the suggestions and comments made by you and by Dr. Simon (Jean-Baptiste) Combes. In what follows, we provide a point-by-point response to your comments and concerns, indicating the location where the changes are placed in the revised manuscript. In order to facilitate the follow-up of these changes, we have included line numbers in the revised manuscript with track changes.

We hope that the revised version fulfils your expectations and that the new structure and organization of the paper is now clearer.

Sincerely,

The authors

Comments from the Academic Editor

Comment 1: 

Response: 

 Thank you. We reviewed the reference list. It is now correct. It was complete, however a link within the text was not working (line 105, page 4) and this have been fixed. 

Comments from Reviewer 2

Comment 1: Line 320 page 17: "About variable age, it has a" => "Age has a" ?

Response: 

 Change made. Thank you.

Comment 2: 

 Page 2 line 36-39: “On the other hand, most of the research is exclusively based on mean regression models analyzing the population’s average income or the expected probability of being poor, disregarding the potential inequalities across income levels. These inequalities justify the need to focus on the poorest.” => proposal : “On the other hand, most of the research is exclusively based on mean regression models analyzing the population’s average income or the expected probability of being poor, disregarding specific effects in poorer income levels. We propose to fill this gap by focusing on the poorest” <= I think this is what the paper does and the "potential inequalities" is a bit vague. I understand that the author may have try to go slowly in the introduction and to not give away too much too early, but the word "inequality" may mislead a reader as "inequality" is nearly quite a domain, I do not think the paper is actually in this domain it is in the literature on poverty (which is linked to inequality, do not get me wrong). If the paper is on inequalities I would expect that the paper looks at rich and poor while it specifically look at poor. I can change my mind if the authors bring some literature explaining carefully the link between poverty and inequalities (I am not saying it is not linked just that it is not that straightforward).

Response: 

 We agree with the reviewer´s assessment regarding the use of the term inequality. Accordingly, the corresponding paragraph in the revised manuscript reads as suggested by the reviewer. Thank you for pointing this out.

Comment 3: 

 Table 2, some cells are gray with nothing in it, is that on purpose? (eg. second row first column) how can be estimated the difference between men and women if the parameter is not estimated?

Response: 

 Thank you for this comment. Regarding the empty cells in Table 2, we have included the following clarification “Empty cells indicate that the corresponding factor is not selected as stable for that specific model, and therefore their coefficient is set to zero” in lines 247 and 2249. In this way, particularly for the case of empty cells in gray, for example for the social networks’ coefficient, this indicates that for women-headed households this is not a relevant factor (coefficient equal to zero), but for men-headed households it is relevant and statistically different than zero. Concerning the second question, as described in pages 12 and 13, about the 3-step strategy applied to overcome the model high-dimensionality, by applying the boosting algorithm a first variable selection is made, however, this selection may include some falsely selected variables. To avoid the erroneous selection of non-relevant factors we apply stability selection. The result of the application of these two steps (boosting and stability selection) is the estimation of a parsimonious model consisting exclusively of stable factors at their most suitable form (i.e. only non-zero regression coefficients), and therefore the coefficient for non-selected factors is set to zero. Setting the coefficients to zero is key since it enables the variable selection and model choice processes (to clarify this in the document, we have included this explanation in the revised manuscript, see lines 201-205 in page 12). It is important to remark that at this point of the strategy there is not an estimation of the corresponding standard errors for the stable selected factors, because confidence bands cannot be derived in the boosting estimation context. This is why we additionally conducted the pointwise bootstrap confidence intervals. These intervals are calculated only for the subset of effects selected as stable in step 2. 

Comment 4: 

 Line 254 on page 15: "For extremely poor families, the coefficient for households without access to credit cards and with a woman as the head is -0.114" => can you tell us what is this parameter? it increase the income to poverty ratio by 11%? Maybe tell the reader that people with credit card are less poor? is that what it means? If I am mistaken then this comment should be taken as evidence that the paragraph is not very clear.

Response: 

 Thank you for pointing this out. We have reformulated this paragraph (lines 278-284 of the revised version with tracked changes) to make it clearer. The revised text reads as follows: “The results indicate that having a household member that holds a credit card is consistently linked to a greater income-to-poverty ratio in rural families. For extremely poor families, not having access to credit cards reduces their income-to-poverty ratio by 0.114 units (around 2.85 times their standard deviation) in women-headed households, and by 0.13 units by 0.13 units (more than three standard deviations) for their male-headed counterparts. The magnitude of the effect significantly varies across poverty levels, but only for men-headed households (see confidence intervals in Table 2). The estimated parameter for the effect of not holding a credit card on the ratio of income-to-poverty of poor families is -0.224 (2.39 standard deviations) for women-headed households and -0.25 (2.55 standard deviations) for those headed by a man” 

In addition, we included a new paragraph about the interpretation of results in the context of quantile models and an example from Table 2 (lines 217-233): “These coefficients indicate the magnitude of the effect that each factor have on the income-to-poverty ratio. It is important to keep in mind that this ratio measures how far or close is a family to live in poverty based on their income, and therefore coefficients can be seen as estimates of the size of the effect as a proportion of the poverty line, i.e. as a share of the income required to cover the cost of the household basic food basket and the non-food basket. Interpretation of the coefficients in the quantile regression model context is basically the same as in other traditional approaches. For categorical covariates, parameters indicate the difference in the estimated effect of a category on the income-to-poverty ratio with respect to the corresponding effect of the reference category. By way of example, when examining the estimated coefficients of women-headed households living in extreme poverty, results indicate that families without access to credit cards have an income-to-poverty ratio that is smaller by 0.114 units in comparison to their counterparts having credit cards. For continuous covariates with purely linear effects, the parameter indicates the change in the income-to-poverty ratio per unit change in the continuous covariate. For instance, for men-headed households living in extreme poverty, an increase of one year in the head’s age decreases the income-to-poverty ratio by 0.003 units. For continuous covariates with nonlinear effects, interpretation is best done by visualizing the corresponding figures. By comparing the estimated coefficients from each of the models, they provide a clear picture of how the effect of the covariates varies across the poverty spectrum and according to the sex of the head.”

Comment 5: 

 Complementary to my previous comment, I think the authors should translate the parameters into meaningful effects, are parameters percentages? increases in standard deviation? at the end of the result section, page 21, line 418, it seems that the effect is an increase in the ratio, is that the case ? Then even this would not be very informative, could it be that the parameter shows the effect as the number of standard deviation of the dependent variable? With OLS it would necessitate to divide the dependent by its own standard deviation and then the parameter would be for a binarry variable like sex, being a female (if male the ref) increase by Beta (eg Beta being positive) standard deviation the dependent variable. If beta is above one then it increase by more than a standard variation , if below it increases by less than one standard variation. But I may not understand well enough the quantile regression.

Response: 

 Thank you. The estimated effects indicate the change in the income-to-poverty ratio, calculated by dividing household income by the poverty threshold (1715.57 MXN). Along the document (lines 43-45, 111-113, 217-233) we included comments about the response variable to better explain it and to explain the interpretation of coefficients. 

We agree that there are different approaches to study poverty, thus we included comments on the advantages of this approach (lines 146-149) and a comment calling for using different indicators of poverty in further research, and the analysis of other distributional parameters of the response, such as the scale and shape (lines 610-613).

Regarding the interpretation in terms of its standard deviation, we would like to comment the following. First, our response variable, income-to-poverty ratio, is not standardized. One advantage of the quantile regression is that this is has an inherent distribution-free character since its estimation is only influenced by the local behavior of the distribution of the response, in this case the income-to-poverty ratio. We decided to analyze its distribution without reshaping nor rescaling it, but in unit terms of the ratio. Therefore, after estimating the parameters, we obtained the standard deviation (we added summary statistics in S1 Table in the supporting information section) and by dividing the parameter by the corresponding standard deviation, we can infer the importance of the estimated effects. Hence, in the results section of the revised manuscript, we commented the estimations in terms of their standard deviation to inform about the size of the effects (see pages 13-22). Thank you for this suggestion.

Comment 6: 

 line 455, page 22 When citing the world bank report on education, a little bit of the explanation of why higher education reduces fertility would be good, so could you add someting like education reduces fertility because it increases contraception and age at marriage?

Response: 

 Thank you. We have reformulated the text to make it clearer (lines 489-495). The updated text reads as follows: “Not holding the mandatory educational level (education lag) is another aspect of multidimensional poverty associated with income for rural poor and extremely poor families. Although more detailed statistical information is required to inquire into the reasons for this result, as stated in [49], a potential explanation could be that completing compulsory education increases income because this education level is associated with a fertility rate reduction and a health status improvement through an increase in contraceptive use and a delay of the age of marriage (or age of first pregnancy), which in turn is linked to a higher productivity.”

---

## [Editor Report · Decision Letter 2]

15 Oct 2021

Examining gender inequalities in factors associated with income poverty in Mexican rural households

PONE-D-21-04773R2

Dear Dr. Juan Armando Torres Munguía

We’re pleased to inform you that your manuscript has been judged scientifically suitable for publication and will be formally accepted for publication once it meets all outstanding technical requirements.

Kind regards,

Carlos Alberto Zúniga-González, Ph.D

Academic Editor

PLOS ONE

Additional Editor Comments (optional):

I suggest that table 2 be transferred to the annexes according to the style rules.
---

## [Editor Report · Acceptance letter]

21 Oct 2021

PONE-D-21-04773R2 

Examining gender inequalities in factors associated with income poverty in Mexican rural households 

Dear Dr. Torres Munguía:

I'm pleased to inform you that your manuscript has been deemed suitable for publication in PLOS ONE. Congratulations! Your manuscript is now with our production department. 

Kind regards, 

on behalf of

Dr. Prof. Carlos Alberto Zúniga-González 

Academic Editor

PLOS ONE